# GradPCA: Leveraging NTK Alignment for Reliable Out-of-Distribution Detection

**Mariia Seleznova**[1*]  **Hung-Hsu Chou**[2]  **Claudio Mayrink Verdun**[3]  **Gitta Kutyniok**[1,4,5,6]

[1]Ludwig-Maximilians-Universität München   [2]University of Pittsburgh   [3]Harvard University
[4]University of Tromso  [5]DLR-German Aerospace Center  [6]Munich Center for Machine Learning

## Abstract

We introduce **GradPCA**, an Out-of-Distribution (OOD) detection method that exploits the low-rank structure of neural network gradients induced by Neural Tangent Kernel (NTK) alignment. GradPCA applies Principal Component Analysis (PCA) to gradient class-means, achieving more consistent performance than existing methods across standard image classification benchmarks. We provide a theoretical perspective on spectral OOD detection in neural networks to support GradPCA, highlighting feature-space properties that enable effective detection and naturally emerge from NTK alignment. Our analysis further reveals that feature quality—particularly the use of pretrained versus non-pretrained representations—plays a crucial role in determining which detectors will succeed. Extensive experiments validate the strong performance of GradPCA, and our theoretical framework offers guidance for designing more principled spectral OOD detectors.

## 1 Introduction

In modern deep learning, models often produce confident but incorrect predictions when presented with inputs outside their training distribution (Goodfellow et al., 2015; Kurakin et al., 2017; Nguyen et al., 2015). Out-of-Distribution (OOD) detection provides a mechanism for models to *know when they don't know*, enabling them to reject inputs beyond their domain of competence. As deep learning systems become integrated into critical decision-making processes, robust OOD detection is essential for ensuring safety and effective human oversight (Amodei et al., 2016).

Although designed to improve the reliability of deep learning, OOD detection methods have themselves proven unreliable (Tajwar et al., 2021; Szyc et al., 2023). Their performance often hinges on subtle assumptions about the model, data, and training procedure, with little guidance as to when these assumptions hold. As a result, OOD detection effectiveness is assessed through empirical validation alone, making the performance difficult to predict and dependent on ad hoc tuning. In response to these challenges, we propose a new OOD detector that combines classical spectral analysis with recent insights into deep learning theory, offering a principled and interpretable approach.

Our main contributions are as follows:

- **GradPCA:** Our method leverages the low-rank structure of gradients in well-trained neural networks (NNs), induced by the *Neural Tangent Kernel (NTK) alignment* phenomenon (Atanasov et al., 2022; Baratin et al., 2021; Seleznova & Kutyniok, 2022; Shan & Bordelon, 2021). Under NTK alignment, gradients of in-distribution (ID) inputs concentrate within stable, low-dimensional subspaces spanned by class-specific directions. GradPCA performs PCA on gradient class-means to efficiently model this subspace and flag inputs whose gradients fall outside it. GradPCA is the first OOD detector to exploit NTK alignment, and its principled design ensures robust performance across realistic detection scenarios.

- **Revisiting spectral OOD detection:** To explain GradPCA's empirical success, we present a theoretical framework for spectral OOD detection in neural networks (NNs). Our framework extends the principles of classical and kernel PCA (Hotelling, 1933; Schölkopf et al., 1998) and allows to derive *one-sided, per-sample OOD certificates* for spectral detectors—a theoretical guarantee rarely found in the predominantly empirical OOD detection literature.

---

[*]Correspondence to: Mariia Seleznova (`selez@math.lmu.de`).

- **Feature quality matters for OOD detection:** We show that the performance of OOD detectors critically depends on the quality of feature representations—specifically, whether they come from general-purpose (pretrained) or task-specific (non-pretrained) models. To demonstrate this, we evaluate existing OOD detectors on a carefully designed benchmark comprising pretrained and non-pretrained models with matched ID accuracy. Notably, methods that detect abnormal model behavior (e.g., confidence-based scores) often *worsen* with improved feature quality, while approaches that exploit geometric regularities in feature space *improve*. This underexplored factor helps reconcile inconsistencies in prior works and highlights the need to account for feature quality when designing future OOD detectors.

- **Empirical validation and open-source implementation:** We evaluate GradPCA exclusively on *publicly available models* and *community-released datasets*, avoiding manual subset selection or ad hoc model training. This principled setup mitigates common evaluation biases and supports fair comparison across diverse settings. We benchmark against widely used competitive baselines spanning diverse methodological approaches, including recent gradient-based detectors, and find that GradPCA delivers the most consistent performance, achieving near state-of-the-art results across all benchmarks. All code and experimental configurations are available at our GitHub repository.

**Notation and outline.** We denote vectors and matrices by bold lowercase and uppercase letters, respectively (e.g., $\mathbf{w}$ for a vector and $\mathbf{F}$ for a matrix). Bold numerals such as $\mathbf{0}$ and $\mathbf{1}$ represent vectors of zeros and ones. The indicator function $\mathbb{1}_A(x)$ equals 1 if $x \in A$ and 0 otherwise. We denote the in-distribution and out-distribution probability density functions by $d\mu_{\mathrm{id}}$ and $d\mu_{\mathrm{ood}}$, with corresponding measures $\mu_{\mathrm{id}}$ and $\mu_{\mathrm{ood}}$. The support of $\mu_{\mathrm{id}}$ is defined as $\mathrm{supp}(\mu_{\mathrm{id}}) := \{x : d\mu_{\mathrm{id}}(x) > 0\}$.

We present the OOD detection problem setup in Section 2, GradPCA and key empirical insights in Section 3, theory in Section 4, experimental setup in Section 5, and conclude with Section 6.

## 2 OOD Detection: Problem Setup and Methods

Before presenting our method, we formulate the OOD detection problem and introduce our categorization of existing OOD detection approaches.

**Problem formulation.** The goal of OOD detection is to construct a detector $D : \mathcal{X} \to \{0, 1\}$ that returns $D(x) = 0$ if $x \sim \mu_{\mathrm{id}}$ and $D(x) = 1$ if $x \sim \mu_{\mathrm{ood}}$, for arbitrary $\mu_{\mathrm{ood}}$. However, this problem is fundamentally ill-posed: no detector is universally optimal across all possible $\mu_{\mathrm{ood}}$ (see Appendix B.1). To make the problem well-posed, we reformulate it as *support recovery*: identifying the support of $\mu_{\mathrm{id}}$, denoted $\mathrm{supp}(\mu_{\mathrm{id}})$. In this setting, the optimal detector is $D^* := \mathbb{1}_{\mathcal{X} \setminus \mathrm{supp}(\mu_{\mathrm{id}})}$, which flags any input outside the support of $\mu_{\mathrm{id}}$ as OOD. We adopt this support-based perspective throughout and refer to any $x \notin \mathrm{supp}(\mu_{\mathrm{id}})$ as *guaranteed* OOD—independent of $\mu_{\mathrm{ood}}$. This view aligns with practice: an input is effectively ID only if the model can correctly classify it, reinforcing that OOD detection should focus on the boundaries of reliable model behavior rather than distributional likelihoods.

**Taxonomy of OOD detectors.** The vast body of work on OOD detection makes it challenging to precisely categorize existing approaches. However, for the purposes of this paper, we find it helpful to broadly classify existing methods into two categories based on their main *assumptions*:

- **Regularity-based:** These methods assume that ID data occupies a compact or structured region in a suitable *feature space*. A sample is flagged as OOD if it deviates significantly from this "regular" region. Classical approaches such as density estimation and (kernel) PCA fall into this category, as do neural OOD detection methods such as Mahalanobis (Lee et al., 2018) and KNN (Sun et al., 2022b). Our method, GradPCA, also belongs to this class.

- **Abnormality-based:** These methods instead focus on identifying "abnormality"—patterns or behaviors that are characteristic of OOD inputs. Rather than relying on proximity to ID samples in any well-defined space, they exploit cues such as predictive uncertainty (e.g., ODIN (Liang et al., 2018), Energy (Liu et al., 2020)) or atypical activation patterns (e.g., ReAct (Sun et al., 2021), GAIA (Chen et al., 2023)).

Our experiments reveal that each class of methods performs best under different conditions (see Section 3.3). In addition to this taxonomy, we include a detailed review of related work in Appendix A.

**Spectral methods: PCA and Kernel PCA.** We focus on *spectral* OOD detectors—a subset of the regularity-based category—since GradPCA belongs to this class. A canonical example of spectral methods is *kernel PCA*, which operates in two steps:

1. Choose a *feature map* $h : \mathcal{X} \to \mathcal{H}$ that embeds the data into a suitable Hilbert space $\mathcal{H}$.
2. Perform standard PCA in the space $\mathcal{H}$.

The challenge lies in step (1): selecting an appropriate $h$ such that ID and OOD samples become separable in the induced space. Classical kernel PCA specifies a kernel function rather than an explicit feature map, but this merely shifts the difficulty to kernel design, retaining the full complexity of representation selection. Prior work on kernel PCA for OOD detection largely relied on generic or ad hoc kernels applied to network activations (Guan et al., 2023; Fang et al., 2024b; Hoffmann, 2007; Xiao et al., 2013). Several recent works have also applied variants of spectral methods to gradients (Wu et al., 2024a; Behpour et al., 2023). GradPCA adds to this line of works by employing the Neural Tangent Kernel (NTK)—a task- and model-specific kernel—and introducing an efficient procedure for performing PCA in gradient space. In the next section, we formalize GradPCA and show that it can be viewed as an approximate kernel PCA method using the NTK kernel.

## 3 GRADPCA

The core idea of GradPCA is to apply classical PCA in the space of neural network's gradients. While conceptually simple, this approach is not directly feasible for modern NNs due to the prohibitive size of the gradient covariance matrix. Consider a NN with output function $f : \mathcal{X} \to \mathbb{R}$ and parameters $\mathbf{w} \in \mathbb{R}^P$. For a dataset $\{(x_i, y_i)\}_{i=1}^N$, the empirical covariance matrix of gradients is given by:

$$\widehat{\mathbf{S}} := \mathbf{F}\mathbf{F}^\top \in \mathbb{R}^{P \times P}, \quad \text{where} \quad \mathbf{F}_{ij} := \nabla_{\mathbf{w}_i} f(x_j). \tag{1}$$

In PCA, the standard approach for handling high-dimensional features is to compute the eigendecomposition of the smaller *dual matrix* $\mathbf{F}^\top \mathbf{F} \in \mathbb{R}^{N \times N}$. However, in modern deep networks, both $P$ (number of parameters) and $N$ (dataset size) are typically too large for this to be practical. To address this, GradPCA exploits the inherent low-rank structure of the gradients induced by *NTK alignment*.

### 3.1 NTK ALIGNMENT

A central observation in our approach is that the dual matrix $\mathbf{F}^\top \mathbf{F} \in \mathbb{R}^{N \times N}$ introduced above is exactly the empirical Neural Tangent Kernel (NTK) evaluated on the dataset:

$$\widehat{\Theta} = \mathbf{F}^\top \mathbf{F}, \quad \text{where} \quad \widehat{\Theta}_{ij} := \langle \nabla_{\mathbf{w}} f(x_i), \nabla_{\mathbf{w}} f(x_j) \rangle. \tag{2}$$

This implies that PCA in gradient space hinges on spectral analysis of the NTK—the kernel that captures correlations between input samples in network's training dynamics.

A growing line of recent research shows that, during training, the empirical NTK of well-performing neural networks progressively aligns with the structure of the learning task—a phenomenon termed *NTK alignment* (Atanasov et al., 2022; Baratin et al., 2021; Shan & Bordelon, 2021). This alignment reshapes the kernel spectrum in a task-relevant manner and is associated with stable training and strong generalization (Chen et al., 2020; Atanasov et al., 2022; Seleznova et al., 2023). This echoes classical kernel methods theory, where kernel–target alignment is a known predictor of generalization (Cristianini et al., 2001; Gönen & Alpaydın, 2011; Canatar et al., 2021).

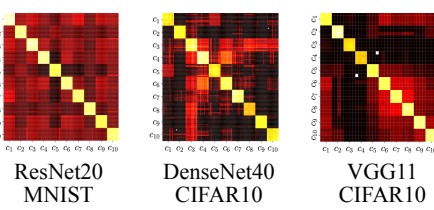

ResNet20 MNIST    DenseNet40 CIFAR10    VGG11 CIFAR10

Figure 1: **Block-diagonal NTK structure.** Heatmaps show the NTK matrix on ID data subset. Lighter colors indicate larger entries. Diagonal blocks correspond to samples from the same class. A detailed description is provided in Appendix E.10.

In classification problems, NTK alignment manifests as an approximate *block-diagonal structure* in the NTK matrix: inputs from the same class exhibit strong correlations, while cross-class interactions are weak (Seleznova et al., 2023). This behavior is deeply related to the principle of *local elasticity* (He & Su, 2020)—where, during training, each

input primarily influences inputs belonging to the same class. As a result, gradients concentrate in a low-dimensional subspace *spanned by class-specific directions*, inducing a pronounced low-rank structure. This structure is visualized for several neural network classifiers in Figure 1. More formally, for a classification problem with $C$ balanced classes, each containing $m := N/C$ samples, the NTK takes the following form:

$$\widehat{\Theta} = \mathbf{G}^\top \mathbf{G} \otimes \mathbf{1}_m \mathbf{1}_m^\top + \boldsymbol{\xi}, \quad \mathbf{G} := [\mathbf{g}^1, \ldots, \mathbf{g}^C], \quad \mathbf{g}^k := \frac{1}{m} \sum_{x_i \in \text{class } k} \nabla_{\mathbf{w}} f(x_i), \tag{3}$$

where $\otimes$ denotes the Kronecker product. The leading term is rank-$C$ (typically $C \ll N, P$), and $\boldsymbol{\xi}$ captures deviations from the perfect alignment. When alignment is strong, $\|\boldsymbol{\xi}\| \leq \epsilon$ for some small $\epsilon$, and the NTK spectrum is dominated by the block-structured term. This approximation underpins our method, and we analyze its robustness with respect to the residual term $\boldsymbol{\xi}$ in Section 4.2.

## 3.2 ALGORITHM

The final observation underlying GradPCA is that the eigendecomposition of the block-structured matrix $\mathbf{G}^\top \mathbf{G} \otimes \mathbf{1}_m \mathbf{1}_m^\top$ depends solely on the class-mean gradients $\mathbf{g}^1, \ldots, \mathbf{g}^C$. As a result, there is no need to store or process the full dataset (or even a large batch) to approximate the principal components of the gradient covariance. It suffices to compute—or approximate—the $C$ class-mean gradient vectors, which greatly reduces the computational cost. The resulting method reduces to performing PCA on a modestly-sized matrix and is summarized in Algorithm 1.

---

**Algorithm 1:** GRADPCA

**Input** : Test input $x$, output function $f : \mathcal{X} \to \mathbb{R}$, ID dataset $\mathcal{D}$, threshold $\delta$, number of PCs $k$
**Output :** Detector decision $D(x) \in \{0, 1\}$

1 **Offline Stage (Training)**
2   Compute class-mean gradients $\mathbf{g}^1, \ldots, \mathbf{g}^C$ and global mean $\bar{\mathbf{g}}$;
3   Form centered matrix $\bar{\mathbf{G}} := [\mathbf{g}^1 - \bar{\mathbf{g}}, \ldots, \mathbf{g}^C - \bar{\mathbf{g}}]$;
4   Compute eigendecomposition of $\bar{\Theta} := \bar{\mathbf{G}}^\top \bar{\mathbf{G}} \in \mathbb{R}^{C \times C}$ via $\bar{\Theta} = \mathbf{V} \Sigma \mathbf{V}^\top$;
5   Compute PCs of $\bar{\mathbf{S}} := \bar{\mathbf{G}} \bar{\mathbf{G}}^\top \in \mathbb{R}^{P \times P}$ via $\mathbf{U}_k = \bar{\mathbf{G}} \mathbf{V}_k \Sigma_k^{-1/2}$;
6   Construct projection matrix $\mathcal{P} := \mathbf{U}_k \mathbf{U}_k^\top$ onto the span of PCs of $\bar{\mathbf{S}}$;
7 **Online Stage (Inference)**
8   Compute centered gradient $\bar{\mathbf{g}}(x) := \nabla_{\mathbf{w}} f(x) - \bar{\mathbf{g}}$;
9   Compute score $s(x) := \|\mathcal{P}\bar{\mathbf{g}}(x)\|/\|\bar{\mathbf{g}}(x)\|$;
10   **return** $D(x) := \mathbb{1}_{[0,\delta)}(s(x))$

---

**Remark 3.1** (Computation of PCs). Since $\bar{\Theta} \in \mathbb{R}^{C \times C}$ and $\bar{\mathbf{S}} \in \mathbb{R}^{P \times P}$ share the same nonzero eigenvalues, PCA can be performed in the smaller $C$-dimensional space. We compute the eigende-composition

$$\bar{\Theta} = \mathbf{V} \Sigma \mathbf{V}^\top, \tag{4}$$

where $\mathbf{V} = [v_1, \ldots, v_C]$ and $\Sigma = \text{diag}(\sigma_1, \ldots, \sigma_C)$ with $\sigma_1 \geq \cdots \geq \sigma_C \geq 0$, and keep the top $k$ eigenpairs by setting $\mathbf{V}_k = [v_1, \ldots, v_k]$ and $\Sigma_k = \text{diag}(\sigma_1, \ldots, \sigma_k)$. The lifted PCs and the associated projection matrix are then given by

$$\mathbf{U}_k := \bar{\mathbf{G}} \mathbf{V}_k \Sigma_k^{-1/2}, \qquad \mathcal{P} := \mathbf{U}_k \mathbf{U}_k^\top. \tag{5}$$

The resulting columns of $\mathbf{U}_k$ are precisely the top $k$ principcal components of $\bar{\mathbf{S}}$.

**Remark 3.2** (Score function). We use the fraction of the centered gradient norm preserved by the principal subspace as the OOD detector's score $s(x) := \|\mathcal{P}\bar{\mathbf{g}}(x)\|/\|\bar{\mathbf{g}}(x)\|$, where ID inputs typically yield larger values. A classical PCA-based detector would instead use the reconstruction error $\|\bar{\mathbf{g}}(x) - \mathcal{P}\bar{\mathbf{g}}(x)\|$. However, recent work (Guan et al., 2023) has shown that the *angle* between a vector and its projection is often more informative for OOD detection than the magnitude of the residual. Our score is precisely an angle-based measure, since $s(x) = \cos(\angle(\bar{\mathbf{g}}(x), \mathcal{P}\bar{\mathbf{g}}(x)))$. We provide a formal justification for why scores of this type, as well as spectral methods more broadly, are principled and effective for OOD detection in Section 4.

**Remark 3.3** (Aggregation). An important practical consideration is that our method is formulated for scalar-valued output functions, whereas standard classifiers produce vector-valued outputs in $\mathbb{R}^C$. As a result, GradPCA requires an *aggregation* step to combine information across output dimensions. In the default version of our implementation, we use the maximum of the network's logits as the scalar output, i.e., $f(x) = \max_{c \in \{1,\dots,C\}} f^c(x)$, where $f^c(x), c \in \{1,\dots,C\}$ are the logits on input $x$.

**Remark 3.4** (Using a subset of parameters.). To further reduce computational cost, we compute gradients with respect to only a subset of parameters—particularly useful when the number of classes $C$ or parameters $P$ is large. Importantly, GradPCA is flexible: it can operate on arbitrary parameter subsets and is not tied to any specific layer or parameter group. Moreover, our ablations in Appendix E.4 show that different parameter subsets may be optimal for different models, likely reflecting which layers carry the most information for OOD detection. Additional implementation details are provided in Section 5 and Appendix C.

### 3.3 GRADPCA PERFORMANCE: CONSISTENCY AND EFFECTS OF FEATURE QUALITY

Before detailing our experimental setup in Section 5, we first highlight the main empirical insights.

**Consistency.** A central observation in our experiments is that OOD detectors' performance can be *highly inconsistent*—even across models with the same architecture and ID dataset—an issue noted in prior work (Tajwar et al., 2021; Szyc et al., 2023). To address this, our evaluation emphasizes consistency by comparing methods across diverse benchmarks designed to reveal both strengths and weaknesses of each approach. As shown in Figure 2, GradPCA achieves the highest average performance among competitive baselines, ranking within the top three methods in virtually every setting. This demonstrates stability of GradPCA's performance, in contrast to the high variability observed for many baselines. We attribute this robustness to the principled foundation of our approach: since NTK alignment is pervasive in well-trained networks, and is tied to strong performance (Chen et al., 2020; Shan & Bordelon, 2021; Seleznova et al., 2023), GradPCA is expected to generalize well across a broad range of realistic settings. Beyond consistency with respect to architectural and dataset choices, prior work has shown that OOD detection can also be surprisingly sensitive to the random seed used during training (Fang et al., 2024a). We examine this factor in Appendix E.3 and find that GradPCA remains stable across multiple training runs with different seeds.

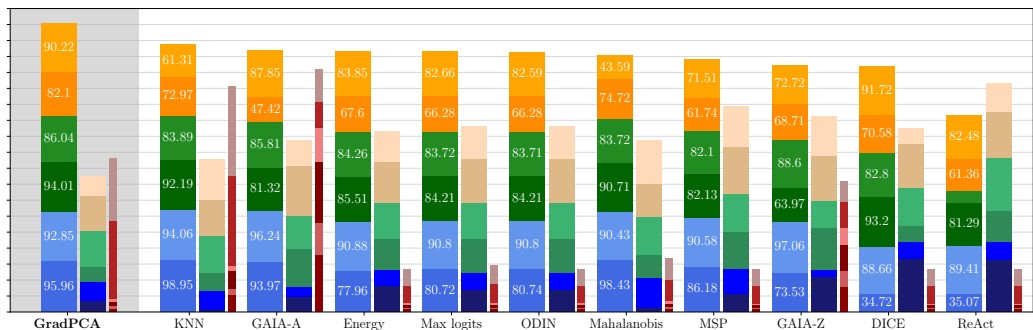

Figure 2: Performance comparison of OOD detection methods across multiple settings. For each method, the left bar shows the stacked average **AUC scores** ↑ on 6 benchmarks described in Section 5 (in order from bottom to top): (1) CIFAR-10 BiT-M (pretrained, Table 4), (2) CIFAR-10 TIMM (Table 4), (3) CIFAR-100 BiT-M (pretrained, Table 1), (4) CIFAR-100 TIMM (Table 1), (5) ImageNet BiT-M (pretrained, Table 2), (6) ImageNet BiT-S (Table 2). The middle bar shows the stacked average **FPR95 scores** ↓ for each method. The right bar shows the stacked **runtime** per sample estimates (Table 3). The methods are ordered left to right by the average AUC score.

**Effects of feature quality.** While prior work has shown that OOD detection performance can vary significantly due to subtle factors such as minor architecture details, random seeds, or dataset splits (Szyc et al., 2023), there is little guidance on how to choose or adapt methods accordingly. We advance this discussion by identifying *feature quality*—whether features come from general-purpose (pretrained) or task-specific (non-pretrained) models—as a key factor for OOD detection performance. We further explain its role through our categorization of OOD detectors into *regularity-based* and

*abnormality-based* (see Section 2). Regularity-based methods perform best with general-purpose features of pretrained models, while abnormality-based methods are more effective when models are trained from scratch, likely because general-purpose features suppress the irregularities these methods aim to detect. Our benchmarks are designed to reflect this: for each ID dataset, we evaluate on two models with similar architectures and ID accuracy but differing feature quality—one pretrained on a large general dataset and fine-tuned on the ID dataset, the other trained directly on the ID dataset. Our results (Figure 2) show that regularity-based methods (GradPCA, KNN (Sun et al., 2022b), Mahalanobis (Lee et al., 2018)) tend to excel on pretrained models, while abnormality-based methods (GAIA (Chen et al., 2023), ODIN (Liang et al., 2018), Energy (Liu et al., 2020)) are closer to state-of-the-art in the non-pretrained settings. This observation offers practical guidance: regularity-based methods are preferable when strong, pretrained features are available, while abnormality-based methods may be more suitable in lower-quality or non-pretrained regimes.

**Computational cost.** GradPCA achieves competitive inference-time efficiency through its paral-lelized implementation and support for batch evaluation, performing on par with fast logits-based methods such as MSP and ODIN on CIFAR (see Figure 2). As with other regularity-based methods, it incurs two additional computational costs: memory usage (requiring storage of $O(C)$ vectors) and a training phase. On modern hardware, GradPCA remains practical on ImageNet, processing over 100 samples per second in our setup (see Appendix D), and provides favorable trade-offs in applications prioritizing robustness and interpretability.

## 4 SPECTRAL OOD DETECTION FOR NEURAL NETWORKS

Spectral methods for OOD detection apply PCA to representations defined by a feature map $h: \mathcal{X} \rightarrow \mathcal{H}$, which embeds inputs into a Hilbert space where ID data exhibit meaningful spectral structure. The effectiveness of such methods depends critically on the choice of $h$. In this section, we formalize the principles underlying spectral OOD detection.

### 4.1 SUFFICIENT CONDITION FOR SPECTRAL OOD DETECTION

We begin with a simple yet broadly applicable result showing how the structure of the covariance matrix can be leveraged for OOD detection. Specifically, the following theorem (proven in Appendix B.2) establishes that any point with a component outside the range of the covariance matrix is *guaranteed* to be OOD. This yields *one-sided, per-sample OOD certificates* for spectral detectors.

**Theorem 4.1** (Sufficient condition for spectral OOD detection). *Let $X \sim \mu_{\mathrm{id}}$ and $h : \mathcal{X} \rightarrow \mathbb{R}^P$ be any function in $L^2(\mu_{\mathrm{id}})$. Consider the covariance matrix*

$$\mathbf{S}(h) := \mathbb{E}[h(X)h(X)^\top]. \tag{6}$$

*Let $\mathcal{P}h(x)$ be the orthogonal projection of $h(x)$ onto the range of $\mathbf{S}(h)$. For any $x \in \mathcal{X}$, if $\|\mathcal{P}h(x)\|_2 < \|h(x)\|_2$ and $h$ is continuous at $x$, then $x$ is OOD.*

Although this result resembles the elementary linear algebra fact that ranges of $A$ and $AA^\top$ are equal, which is used in some classical PCA works (Schölkopf et al., 1998; Hastie et al., 2009), our version introduces a probabilistic perspective that, to our knowledge, has not been formally stated in the OOD detection literature. In our view, this provides foundational motivation for the following detector:

$$D_h(x) := \mathbb{1}_{[0,\delta)}(s(x)), \quad s(x) := \frac{\|\mathcal{P}h(x)\|^2}{\|h(x)\|^2}, \tag{7}$$

where $s : \mathcal{X} \rightarrow [0, 1]$ is a *score function* and $\delta \in (0, 1]$ is a threshold. We next show how this connects to the practical use of PCA for OOD detection.

### 4.2 CONNECTION TO PCA AND GRADPCA

We now show how Theorem 4.1 yields a practical OOD certificate for PCA-based detectors when the empirical covariance matrix approximates a low-rank population covariance $\mathbf{S}(h)$.

**Theorem 4.2** (Robust OOD certificate for PCA). *Consider PCA applied to a matrix $\hat{\mathbf{S}} \in \mathbb{R}^{P \times P}$ (e.g., estimated from a noisy sample). Let $h$ and $\mathbf{S}(h)$ be as in Theorem 4.1, and assume the following:*

$$\|\mathbf{S}(h) - \hat{\mathbf{S}}\|_2 \leq \epsilon, \quad \mathrm{rank}(\mathbf{S}(h)) = C, \tag{8}$$

*i.e.,* $\hat{\mathbf{S}}$ *approximates a rank-C covariance matrix* $\mathbf{S}(h)$*. Let* $\hat{\mathcal{P}}_C$ *denote the orthogonal projector onto the top* $C$ *eigenvectors of* $\hat{\mathbf{S}}$*, and let* $\lambda_C$ *be the* $C$*-th largest eigenvalue of* $\mathbf{S}(h)$*. Then, for any input* $x \in \mathcal{X}$*, the following condition is sufficient to guarantee that* $x$ *is OOD:*

$$s_{\text{PCA}}(x) < 1 - \frac{2\epsilon}{\lambda_C - \epsilon}, \quad s_{\text{PCA}}(x) := \frac{\|\hat{\mathcal{P}}_C h(x)\|^2}{\|h(x)\|^2}. \tag{9}$$

The proof, based on standard projection bounds and the Davis–Kahan theorem (Davis & Kahan, 1970), is provided in Appendix B.3. This result characterizes the *robustness* of spectral OOD detection to perturbations in the covariance structure—arising from noise, finite-sample effects, or deviations from an ideal low-rank model (e.g., the residual term $\boldsymbol{\xi}$ in equation 3). This framework aligns naturally with GradPCA, where we approximate the empirical (potentially full-rank) gradient covariance using a low-rank surrogate constructed from class-mean gradient vectors.

### 4.3 Necessary Condition for Spectral OOD Detection

The results discussed so far provide only *sufficient*, but *not necessary*, conditions for identifying a point $x \in \mathcal{X}$ as OOD. Indeed, depending on the choice of $h$, Theorem 4.1 may allow to identify all OOD points or, in the worst case, none at all. We illustrate this by the following two examples.

**Example 4.3** (Best case). *Let* $h : \mathcal{X} \rightarrow \mathbb{R}^P$ *be an indicator function of* $\mu_{\text{id}}$ *support, defined as* $h(x) = \boldsymbol{v} \cdot \mathbb{1}_{\text{supp}(\mu_{\text{id}})}(x)$ *for some fixed nonzero vector* $\boldsymbol{v} \in \mathbb{R}^P$*. Then,* $x \in \mathcal{X}$ *is OOD if and only if* $\|\mathcal{P}h(x)\|_2 < \|h(x)\|_2$*, and detector* $D_h$ *(defined in equation 7) identifies all OOD points.*

**Example 4.4** (Worst case). *Let* $h : \mathcal{X} \rightarrow \mathbb{R}^P$ *be a function such that* $\text{rank}(\mathbf{S}(h)) = P$*. Then* $\mathcal{P}h(x) = h(x)$ *for all* $x \in \mathcal{X}$*, and no OOD points can be detected by* $D_h$*.*

As illustrated by the worst-case example, the following condition is necessary for spectral OOD detection to be effective. A formal proof is provided in Appendix B.5.

**Theorem 4.5** (Necessary condition for spectral OOD detection). *Consider the same setting as in Theorem 4.1, and let* $D_h$ *be the spectral OOD detector defined in equation 7, with an arbitrary threshold* $\delta \in (0, 1]$*. Then, the following condition is necessary for* $D_h$ *to be effective (i.e., have non-zero sensitivity):*

$$\text{rank}(\mathbf{S}(h)) < \dim\left(\{h(x) : x \in \mathcal{X}\}\right). \tag{10}$$

In other words, the image of the OOD data must not lie entirely within the image of the ID data. The ideal setting occurs when the feature map $h$ embeds all ID points into a low-rank subspace while mapping OOD points outside of it—precisely the structure exhibited in the best-case example. In the absence of any knowledge about the distribution or the structure of $h$, it is generally intractable to formulate necessary and sufficient conditions for a given point $x \in \mathcal{X}$ to be OOD.

### 4.4 Choice of Feature Map for Spectral OOD Detection in Neural Networks

Based on the previous discussion, an effective feature map $h$ for spectral OOD detection should satisfy two key principles consistent with standard PCA and kernel PCA intuition: (1) The image of ID data $\{h(x) : x \in \text{supp}(\mu_{\text{id}})\}$ should concentrate in a low-rank subspace; (2) the image of OOD data $\{h(x) : x \notin \text{supp}(\mu_{\text{id}})\}$ should fall outside this subspace. We now evaluate the following feature spaces commonly used in OOD detection methods against these criteria: *logits*, *hidden activations* (hidden-layer features), and *gradients*.

**Logits:** In classification tasks, the logits are a vector in $\mathbb{R}^C$, where $C$ is the number of classes. Since logits are trained to approximate one-hot vectors of class labels, the image of ID data $\{h(x) : x \in \text{supp}(\mu_{\text{id}})\}$ spans the whole $\mathbb{R}^C$. As a result, the logits space lacks the low-rank structure necessary for effective spectral OOD detection. Consistent with this, competitive logits-based OOD detectors (e.g., ODIN (Liang et al., 2018), Energy (Liu et al., 2020)) do not rely on spectral methods.

**Hidden activations:** The spectral structure of hidden-layer features depends heavily on the choice of layer. When $h$ maps to the *penultimate layer* features, the *Neural Collapse* (NC) phenomenon (Papyan et al., 2020) shows that the ID data $\{h(x) : x \in \text{supp}(\mu_{\text{id}})\}$ indeed often concentrates in a subspace of rank $C$, much smaller than the ambient dimension. However, this structure degrades in earlier layers (Parker et al., 2023). Several regularity-based OOD detectors (e.g. Mahalanobis (Lee et al.,

2018) or KNN (Sun et al., 2022b)) operate in the activations space, suggesting that it often retains sufficient structure in practice. In addition, a recent work proposed an OOD detector explicitly based on the NC phenomenon (Liu & Qin, 2025). That said, it remains unclear whether OOD data $\{h(x) : x \notin \text{supp}(\mu_{\text{id}})\}$ consistently deviates from the ID subspace. In fact, results from the *adversarial examples* literature (Carlini & Wagner, 2017) shows that OOD inputs can be crafted to effectively mimic ID features in hidden layers, which limits effectiveness of spectral detectors in hidden activations space.

**Gradients:** As discussed in Section 3, the NTK alignment phenomenon implies that the ID gradients $\{h(x) : x \in \text{supp}(\mu_{\text{id}})\}$ approximately lie in a low-dimensional subspace of rank $C$. Empirically, this low-rank structure is robust across architectures and datasets (Baratin et al., 2021; Atanasov et al., 2022; Shan & Bordelon, 2021; Seleznova et al., 2023). Notably, NTK alignment has been observed to happen more often than NC (Seleznova et al., 2023), and is not limited to the final stage of training (Fort et al., 2020). Unlike forward-pass features, gradients belong to a much higher-dimensional space, making the aligned subspace significantly harder to mimic—e.g., through adversarial examples. This high ambient dimensionality amplifies spectral separation, reducing the likelihood that OOD samples project strongly onto the principal components of the ID-induced subspace.

## 5 EXPERIMENTS

In this section, we outline the technical setup of our experiments. We evaluate on three ID datasets: CIFAR-10 (Table 4), CIFAR-100 (Table 1), and ImageNet-1k (Table 2, Table 6). Aggregated results across the main benchmarks are summarized in Figure 2. To ensure reproducibility and control for variability, we use only *publicly available* datasets and models. Benchmarks for each dataset include at least two models to assess the impact of representations quality (see Section 3.3).

**Implementation of GradPCA.** Our implementation of GradPCA follows Algorithm 1, with two key extensions: (1) a configurable *aggregation scheme* over output heads, and (2) the option to compute gradients with respect to a *subset of trainable parameters*, improving scalability. To truncate the spectrum, we use a configurable threshold $\epsilon$ (default: 0.99), denoting the fraction of the trace retained by the principal components. The default variant, referred to as **GradPCA**, operates on gradients with respect to the last hidden layer parameters, averages logits across output dimensions before PCA application, and computes gradient class-means sequentially over the training data—making it scalable to large datasets. Computational costs are described in Appendix D.

In addition, we implement several *variants* of GradPCA: (1) **GradPCA+DICE**, which incorporates the DICE method (Sun & Li, 2022) by sparsifying gradient class-means before PCA; (2) **GradPCA-Batch**, which computes the empirical NTK on a batch instead of relying on class-means; (3) **GradPCA-Vec**, which omits output aggregation and computes GradPCA separately for each of the $C$ output heads, producing $C$ scores that can be aggregated post hoc. While these variants can be advantageous in certain scenarios, our main evaluation (Figure 2) focuses on the base GradPCA variant due to its efficiency and *lack of hyperparameter tuning*—enabling fair comparison across models and datasets. We include several GradPCA variants in Tables 1 and 2, and extensive ablations in Appendix E. We release implementation of GradPCA in JAX (Bradbury et al., 2018).

**Baselines.** We compare GradPCA against a range of established methods that have demonstrated effectiveness across diverse benchmarks. In particular, we include regularity-based methods such as Mahalanobis (Lee et al., 2018) and Deep $k$-Nearest Neighbors (KNN) (Sun et al., 2022b), which are methodologically aligned with GradPCA and serve as natural points of comparison. We also incorporated a range of confidence-based scores (MSP (Hendrycks & Gimpel, 2017), ODIN (Liang et al., 2018), Energy (Liu et al., 2020)), as well as DICE (Sun & Li, 2022), which leverages sparsity, and ReAct (Sun et al., 2021), which relies on abnormal activation patterns. Collectively, these baselines span a broad spectrum of approaches commonly found in the literature. Since our method operates in the gradient space, we include comparisons with three state-of-the-art gradient-based OOD detectors: GAIA (Chen et al., 2023), GradOrth (Behpour et al., 2023), and the Projected Gradients method of Wu et al. (2024a). We additionally compare against two recent PCA-based approaches: the Revisited PCA method of Guan et al. (2023) and Kernel PCA (CoRP) (Fang et al., 2024b). Finally, we compare against the recent Neural Collapse Inspired (NCI) detector (Liu & Qin, 2025), which similarly exploits low-rank structure (in penultimate features rather than gradients). Implementation and experimental setup details for all baselines are provided in Appendix C.

## 5.1 CIFAR BENCHMARK

| | Methods | Labels | SVHN | | Places | | LSUN-c | | LSUN-r | | iSUN | | Textures | | Average | |
|---|---|---|---|---|---|---|---|---|---|---|---|---|---|---|---|---|
| | | | FPR95↓ | AUROC↑ | FPR95↓ | AUROC↑ | FPR95↓ | AUROC↑ | FPR95↓ | AUROC↑ | FPR95↓ | AUROC↑ | FPR95↓ | AUROC↑ | FPR95↓ | AUROC↑ |
| ResNetV2-50 (BiT-M) | Max logits | | 65.17 | 87.17 | 59.08 | 84.43 | 68.80 | 81.34 | 67.41 | 81.34 | 68.50 | 80.76 | 54.79 | 87.44 | 63.96 | 84.21 |
| | MSP | | 68.42 | 85.87 | 67.05 | 81.20 | 72.56 | 82.93 | 70.21 | 79.95 | 71.32 | 79.47 | 64.33 | 83.35 | 68.98 | 82.13 |
| | ODIN | | 65.18 | 87.17 | 59.11 | 84.43 | 68.86 | 84.11 | 67.43 | 81.34 | 68.51 | 80.76 | 54.79 | 87.44 | 63.98 | 84.21 |
| | Energy | | 63.74 | 87.44 | 47.17 | 87.05 | 65.29 | 84.41 | 67.19 | 81.92 | 66.94 | 81.39 | 40.15 | 90.83 | 58.41 | 85.51 |
| | DICE | | 13.92 | 97.50 | 30.01 | 93.07 | 9.29 | 97.91 | 59.06 | 85.47 | 55.54 | 85.75 | 1.97 | 99.50 | 28.30 | 93.20 |
| | Mahalanobis | ✓ | 38.07 | 93.74 | 27.26 | 94.00 | 27.12 | 95.81 | 81.08 | 82.04 | 82.31 | 78.80 | 0.41 | 99.88 | 42.71 | 90.71 |
| | KNN | ✓ | 14.01±0.02 | 96.80±0.07 | 42.31±1.19 | 88.51±0.12 | 13.09±0.75 | 97.55±0.08 | 59.95±1.21 | 87.32±0.48 | 63.82±1.15 | 83.81±0.71 | 3.46±0.32 | 99.17±0.05 | 32.77±0.81 | 92.19±0.25 |
| | ReAct | | 44.77 | 89.03 | 69.72 | 76.87 | 52.67 | 85.66 | 86.61 | 71.04 | 85.82 | 73.55 | 11.54 | 97.36 | 58.87 | 81.32 |
| | GAIA-A | | 52.74 | 89.12 | 68.06 | 79.66 | 67.25 | 83.95 | 82.79 | 76.53 | 85.82 | 85.13 | 62.11 | 85.13 | 69.80 | 81.32 |
| | GAIA-Z | | 79.76 | 81.78 | 98.01 | 46.36 | 65.26 | 78.59 | 99.63 | 37.74 | 97.32 | 46.73 | 32.19 | 92.62 | 78.69 | 63.97 |
| | GradOrth | | 26.974 | 92.97 | 18.520 | 96.53 | 28.636 | 93.86 | 56.030 | 86.86 | 52.66 | 88.79 | 7.901 | 98.32 | 31.79 | 92.89 |
| | Revisited PCA | | 25.62 | 94.85 | 26.67 | 93.47 | 35.90 | 91.42 | 39.74 | 90.04 | 47.83 | 89.39 | 9.09 | 96.24 | 30.14 | 92.57 |
| | Proj. Grads | ✓ | 36.78 | 93.01 | 56.38 | 85.66 | 58.94 | 87.88 | 58.89 | 85.84 | 61.49 | 84.28 | 38.81 | 90.40 | 51.88 | 87.85 |
| | Kernel PCA (CoRP) | | 6.17 | 98.70 | 13.29 | 97.31 | 5.07 | 98.91 | 55.99 | 89.45 | 58.07 | 87.32 | 0.12 | 99.96 | 23.12 | 95.38 |
| | NCI (w/o filter) | | 76.36 | 87.54 | 72.98 | 82.73 | 62.82 | 85.78 | 76.92 | 82.61 | 88.41 | 80.55 | 40.91 | 91.32 | 69.73 | 85.09 |
| | **GradPCA** | ✓ | 17.20 | 96.58 | 29.64 | 93.56 | 8.28 | 98.42 | 51.75 | 88.93 | 56.93 | 87.34 | 3.41 | 99.24 | 27.87 | 94.01 |
| | **GradPCA+DICE** | ✓ | 18.11 | 96.57 | 30.77 | 93.34 | 8.12 | 98.46 | 55.72 | 87.43 | 59.53 | 86.27 | 3.59 | 99.20 | 29.31 | 93.54 |

| | Methods | Labels | SVHN | | Places | | LSUN-c | | LSUN-r | | iSUN | | Textures | | Average | |
|---|---|---|---|---|---|---|---|---|---|---|---|---|---|---|---|---|
| | | | FPR95↓ | AUROC↑ | FPR95↓ | AUROC↑ | FPR95↓ | AUROC↑ | FPR95↓ | AUROC↑ | FPR95↓ | AUROC↑ | FPR95↓ | AUROC↑ | FPR95↓ | AUROC↑ |
| ResNet-34 (TIMM) | Max logits | | 71.38 | 83.83 | 69.98 | 83.47 | 64.04 | 85.00 | 75.26 | 79.34 | 62.14 | 86.20 | 66.08 | 84.45 | 68.15 | 83.72 |
| | MSP | | 71.47 | 83.40 | 68.21 | 83.60 | 74.64 | 79.78 | 68.92 | 83.35 | 71.69 | 81.83 | 75.44 | 80.66 | 71.73 | 82.10 |
| | ODIN | | 71.38 | 83.83 | 64.04 | 85.00 | 75.26 | 79.34 | 62.14 | 86.20 | 66.08 | 84.45 | 69.98 | 83.47 | 68.15 | 83.71 |
| | Energy | | 73.15 | 83.79 | 61.66 | 85.50 | 77.32 | 79.18 | 56.32 | 87.24 | 61.57 | 85.38 | 65.70 | 84.44 | 65.95 | 84.26 |
| | DICE | | 83.35 | 81.41 | 67.55 | 84.00 | 68.91 | 75.29 | 58.77 | 86.82 | 64.40 | 84.78 | 64.70 | 84.53 | 71.28 | 82.80 |
| | Mahalanobis | ✓ | 64.76 | 87.75 | 72.30 | 84.83 | 63.01 | 86.09 | 80.09 | 81.99 | 79.99 | 81.33 | 78.75 | 80.31 | 73.15 | 83.72 |
| | KNN | | 64.73±0.35 | 87.26±0.07 | 67.77±0.13 | 85.20±0.01 | 67.76±0.11 | 84.55±0.06 | 70.51±0.24 | 83.46±0.03 | 72.46±0.14 | 82.33±0.02 | 74.74±0.21 | 80.54±0.11 | 69.66±0.20 | 83.89±0.05 |
| | ReAct | | 100.00 | 17.69 | 99.94 | 20.66 | 99.37 | 18.59 | 100.00 | 21.24 | 100.00 | 21.66 | 99.79 | 25.87 | 99.85 | 20.95 |
| | GAIA-A | | 49.73 | 92.52 | 64.28 | 86.00 | 61.22 | 89.39 | 63.86 | 84.69 | 67.30 | 83.32 | 69.07 | 78.94 | 62.58 | 85.81 |
| | GAIA-Z | | 13.33 | 97.34 | 13.44 | 97.34 | 85.66 | 76.45 | 28.51 | 94.66 | 88.11 | 80.87 | 78.50 | 84.94 | 51.26 | 88.60 |
| | GradOrth | | 81.79 | 80.57 | 63.20 | 86.82 | 70.88 | 83.66 | 69.73 | 86.82 | 66.97 | 87.16 | 60.48 | 87.98 | 68.84 | 85.50 |
| | Revisited PCA | | 65.01 | 84.27 | 55.08 | 88.20 | 56.25 | 86.79 | 66.46 | 82.87 | 78.11 | 80.87 | 61.28 | 85.10 | 60.40 | 85.60 |
| | Proj. Grads | | 73.53 | 76.70 | 76.01 | 77.03 | 70.51 | 80.08 | 73.00 | 80.28 | 77.13 | 73.01 | 78.15 | 72.19 | 74.72 | 76.55 |
| | Kernel PCA (CoRP) | | 59.17 | 87.36 | 55.72 | 88.04 | 52.61 | 89.18 | 62.77 | 87.09 | 54.69 | 88.60 | 57.77 | 87.63 | 57.78 | 87.65 |
| | NCI (w/o filter) | | 63.05 | 85.46 | 68.07 | 86.37 | 65.39 | 87.79 | 82.05 | 76.91 | 81.16 | 77.13 | 84.09 | 76.87 | 73.97 | 81.76 |
| | **GradPCA** | ✓ | 61.22 | 89.10 | 72.30 | 84.63 | 62.71 | 87.31 | 63.45 | 87.11 | 73.97 | 84.25 | 73.01 | 83.85 | 67.78 | 86.04 |
| | **GradPCA+DICE** | ✓ | 61.21 | 89.10 | 70.24 | 86.10 | 61.27 | 87.84 | 64.81 | 86.62 | 70.59 | 86.40 | 69.95 | 85.79 | 66.34 | 86.98 |

Table 1: **CIFAR-100.** For each architecture and evaluation metric (FPR95, AUROC), the best-performing method is shown in **bold**, and the second and third best methods are underlined. The **Labels** column indicates whether the method requires access to ID data labels during training (✓).

**Datasets:** For CIFAR-10 and CIFAR-100, we use the standard training splits during the training phase and the corresponding test splits as ID data during the detection phase. We evaluate against a range of commonly used OOD datasets: SVHN (Netzer et al., 2011), Places (Zhou et al., 2017), LSUN (Yu et al., 2015a), iSUN (Xu et al., 2015), and Textures (Cimpoi et al., 2014). SVHN (test split), Places (validation), and Textures (full dataset) are taken directly from the predefined splits in the TensorFlow Datasets collection (TFD). For LSUN and iSUN, we rely on curated subsets for OOD detection benchmarks introduced in the ODIN paper (Liang et al., 2018).

**Models:** We consider two publicly available models: (1) ResNetV2-50 from Big Transfer (BiT) repository (Beyer et al., 2022) pretrained on ImageNet-21k (BiT-M) and fine-tuned on CIFAR, (2) ResNet-34 trained directly on CIFAR from PyTorch Image Models (TIMM) (Wightman, 2019).

**Results:** As shown in Table 1, PCA-based methods (including GradPCA, CoRP, and Revisited PCA) as well as other regularity-based approaches such as Mahalanobis and KNN, achieve strong performance on CIFAR-100 when using general-purpose (pretrained) features. However, as discussed in Section 3.3, the landscape shifts substantially in the task-specific (non-pretrained) setting: here, the abnormality-based GAIA method excels, while confidence-based detectors (e.g., ODIN, MSP) also approach state-of-the-art performance. In contrast, most regularity-based methods degrade noticeably in this scenario, a decline not observed among abnormality-based approaches. An additional observation, also visible in Figure 2, is that methods that excel under certain conditions may fail entirely in others due to violations of their core assumptions. Examples in this benchmark include ReAct and GAIA-Z each underperforming in one of the settings.

## 5.2 IMAGENET BENCHMARK

| | Methods | Places | | SUN | | iNaturalist | | Textures | | Average | | Places | | SUN | | iNaturalist | | Textures | | Average | |
|---|---|---|---|---|---|---|---|---|---|---|---|---|---|---|---|---|---|---|---|---|---|---|
| | | FPR95↓ | AUROC↑ | FPR95↓ | AUROC↑ | FPR95↓ | AUROC↑ | FPR95↓ | AUROC↑ | FPR95↓ | AUROC↑ | FPR95↓ | AUROC↑ | FPR95↓ | AUROC↑ | FPR95↓ | AUROC↑ | FPR95↓ | AUROC↑ | FPR95↓ | AUROC↑ |
| ResNetV2-101 (BiT-M) | Max logits | 89.36 | 59.98 | 87.94 | 61.69 | 55.73 | 84.78 | 94.85 | 58.66 | 81.97 | 66.28 | 73.73 | 77.77 | 69.78 | 79.92 | 29.76 | 93.48 | 68.96 | 79.47 | 60.56 | 82.66 |
| | MSP | 92.84 | 57.05 | 92.97 | 56.72 | 71.28 | 76.50 | 95.44 | 56.71 | 88.13 | 61.74 | 84.40 | 67.28 | 82.79 | 67.94 | 52.62 | 83.94 | 83.04 | 66.88 | 75.71 | 71.51 |
| | ODIN | 89.37 | 59.98 | 87.95 | 61.68 | 55.74 | 84.78 | 94.85 | 58.66 | 81.98 | 66.28 | 73.86 | 77.70 | 70.03 | 79.84 | 29.98 | 93.43 | 69.18 | 79.40 | 60.76 | 82.59 |
| | Energy | 86.94 | 60.96 | 83.94 | 63.36 | 46.07 | 87.17 | 94.26 | 58.91 | 77.80 | 67.60 | 70.79 | 78.91 | 66.36 | 81.28 | 25.21 | 94.38 | 64.20 | 80.84 | 56.64 | 83.85 |
| | DICE | 88.34 | 65.87 | 81.67 | 71.39 | 71.53 | 87.40 | 96.00 | 57.65 | 84.38 | 70.58 | 38.55 | 89.18 | 29.67 | 92.15 | 8.46 | 97.73 | 40.77 | 87.81 | 29.36 | 91.72 |
| | KNN | 91.82 | 56.75 | 92.03 | 57.58 | 33.01 | 92.07 | 31.00 | 92.48 | 61.97 | 74.72 | 97.90 | 30.61 | 97.56 | 28.53 | 99.09 | 23.30 | 30.21 | 91.93 | 81.14 | 43.59 |
| | KNN | 90.33 | 56.29 | 92.14 | 57.22 | 37.95 | 90.85 | 47.49 | 89.78 | 67.15 | 72.97 | 92.68 | 51.01 | 92.96 | 51.78 | 93.69 | 48.81 | 24.12 | 93.64 | 75.86 | 61.31 |
| | ReAct | 92.86 | 53.85 | 91.21 | 55.73 | 62.24 | 83.02 | 96.44 | 52.85 | 85.68 | 61.36 | 69.42 | 76.41 | 63.95 | 78.68 | 24.83 | 93.45 | 58.81 | 81.36 | 54.25 | 82.48 |
| | GAIA-A | 95.77 | 47.23 | 94.65 | 51.41 | 85.31 | 60.07 | 98.95 | 30.96 | 93.67 | 47.42 | 53.54 | 85.66 | 46.05 | 89.33 | 21.69 | 95.33 | 69.78 | 81.07 | 47.77 | 87.85 |
| | GAIA-Z | 97.81 | 56.09 | 98.77 | 57.51 | 79.55 | 79.15 | 62.39 | 82.08 | 84.63 | 68.71 | 94.77 | 58.63 | 93.69 | 62.28 | 85.23 | 75.82 | 26.44 | 94.17 | 75.03 | 72.72 |
| | GradOrth | 95.45 | 66.006 | 93.52 | 71.32 | 96.68 | 67.57 | 97.857 | 59.44 | 95.88 | 66.08 | 55.24 | 82.914 | 43.67 | 88.28 | 19.85 | 95.122 | 21.112 | 95.03 | 34.97 | 90.34 |
| | Revisited PCA | 80.76 | 64.12 | 75.78 | 67.98 | 27.89 | 92.11 | 73.30 | 74.69 | 64.43 | 74.89 | 57.32 | 80.54 | 57.32 | 83.68 | 20.05 | 95.33 | 35.55 | 89.91 | 44.83 | 87.36 |
| | Proj. Grads | 84.02 | 72.48 | 82.31 | 75.40 | 57.51 | 83.69 | 91.32 | 62.63 | 78.29 | 73.55 | 71.04 | 75.73 | 66.35 | 78.56 | 32.52 | 89.71 | 73.20 | 68.10 | 60.78 | 78.03 |
| | Kernel PCA (CoRP) | 86.61 | 60.37 | 83.81 | 64.63 | 25.03 | 94.43 | 28.82 | 93.65 | 56.07 | 78.77 | 81.83 | 65.62 | 76.97 | 71.21 | 66.30 | 81.31 | 15.37 | 96.48 | 60.11 | 78.65 |
| | NCI (w/o filter) | 92.41 | 63.75 | 92.41 | 69.69 | 46.84 | 91.50 | 86.36 | 72.52 | 79.50 | 74.37 | 91.14 | 68.52 | 88.61 | 72.93 | 69.62 | 84.63 | 65.91 | 82.90 | 78.82 | 77.25 |
| | **GradPCA** | 83.12 | 71.55 | 75.16 | 78.68 | 45.84 | 91.69 | 60.74 | 86.49 | 66.22 | 82.10 | 61.27 | 81.75 | 49.35 | 87.20 | 17.78 | 95.80 | 18.08 | 96.14 | 36.62 | 90.22 |
| | **GradPCA+DICE** | 76.97 | 75.5i | 64.82 | 82.95 | 37.68 | 93.10 | 44.07 | 90.40 | 55.88 | 85.49 | 65.07 | 80.10 | 53.16 | 85.85 | 19.25 | 95.48 | 18.16 | 96.20 | 38.91 | 89.41 |

Table 2: **ImageNet.** For each architecture and evaluation metric (FPR95, AUROC), the best-performing method is shown in **bold**, and the second and third best methods are underlined.

**Datasets:** We use the full ImageNet-1k training split in the training phase. While a subset could be used, our method is scalable enough to handle the entire dataset efficiently. For the detection phase, we treat ImageNetV2 (Recht et al., 2019) as the ID dataset. As OOD datasets, we use curated subsets of Places (Zhou et al., 2017), SUN (Xiao et al., 2010), and iNaturalist (Van Horn et al., 2018), released by the MOS paper (Huang & Li, 2021) to ensure minimal category overlap with ImageNet. This control is especially important given that these datasets contain many overlapping classes. We additionally include the full Textures dataset (Cimpoi et al., 2014) as an OOD source.

**Models:** We consider two BiT models (Beyer et al., 2022): (1) ResNetV2-101 pretrained on ImageNet-21k (BiT-M) and fine-tuned on ImageNet-1k, (2) ResNetV2-50 trained directly on ImageNet-1k (BiT-S). In addition, we include a benchmark for (3) Vision Transformer (ViT-B/16) (Dosovitskiy et al., 2021) pretrained on ImageNet-21k and fine-tuned on ImageNet-1k in Appendix E.2.

**Results:** Consistent with trends observed on CIFAR, GradPCA achieves competitive performance on ImageNet—ranking first in the pretrained setting and third in the non-pretrained setting, where DICE performs best. GradPCA also ranks first on the ViT-B/16 benchmark (Appendix, Table 6). Most other regularity-based methods lag behind the state of the art on ImageNet benchmarks, with the notable exception of GradORTH, which performs strongly on the BiT-S model but underperforms on the BiT-M model. As with CIFAR, method rankings vary substantially between the two models. Notably, the pretrained ImageNet BiT benchmark is substantially more challenging than the non-pretrained variant for all detectors, despite the models having similar architectures and ID accuracy. This underscores the sensitivity of OOD detection to subtle differences in evaluation setup. We report results for GradPCA+DICE ($p = 0.8$) in Tables 2 and 1, along with ablations and additional GradPCA variants in Appendix E. While certain tuned variants yield modest gains in specific cases, their performance is inconsistent across settings; for this reason, we focus on the default tuning-free GradPCA in our main discussion.

## 6 CONCLUSIONS AND BROAD IMPACT

This paper introduces GradPCA, an OOD detection method based on the low-rank structure of gradients in well-trained NNs, induced by NTK alignment. Our approach bridges recent advances in OOD detection with classical PCA-based intuition, clarifying when spectral methods are applicable in modern deep learning. In addition, our theoretical analysis and the observation about feature quality as a critical factor for OOD detectors' performance offer practical guidance for selecting appropriate detectors based on the training regime. Such guidance remains rare in the OOD detection literature. Finally, our work establishes a novel connection between the NTK—a common topic in deep learning theory—and OOD detection, a largely empirical field with limited theoretical grounding.

**Limitations.** GradPCA is built on the assumption that the NTK structure effectively separates ID and OOD points in gradient space. While this may not hold universally, the assumption is both explicit and empirically testable—unlike many heuristic or implicit assumptions used in prior works. Importantly, this assumption has been found to hold in a wide range of practical settings, including publicly available, well-trained models commonly used in OOD evaluation. A second limitation is memory scalability with respect to the number of classes: GradPCA stores up to $O(C)$ gradient vectors, which can be costly for large $C$. This requirement is common to many regularity-based approaches and can be mitigated through (1) retained variance threshold $\epsilon$ (see Appendix E.7) and (2) parameter subset selection. Our method shares two other limitations with Mahalanobis and prototype-based detectors (see Appendix A). First, it requires ID labels during training, though this can be mitigated using pseudo-labels or clustering (Sehwag et al., 2021). Second, it is inherently tied to classification tasks, where NTK alignment induces the low-rank structure our method relies on.

**Future work.** By establishing a connection between NTK alignment and OOD detection, our work enables future advances in NTK theory to directly inform and improve methods like GradPCA. As the study of feature learning dynamics and NTK structure remains an active area of theoretical research, we anticipate that forthcoming insights in this domain could be leveraged to design more principled and effective OOD detectors. On the other hand, our results may motivate future works on the relationship between NTK alignment and generalization. On the practical side, several directions remain open, including scalability to very large models and extensions beyond classification setting.

**Reproducibility statement.**  We provide full details of our experimental setup in the main text (Section 5) and supplementary materials (Appendix C), and include anonymized source code in the supplementary submission. The code will be made publicly available upon acceptance. All experiments are conducted using publicly available models and datasets. Where applicable, we use official or community-released implementations of baseline methods. Dataset splits follow standard OOD benchmarks, and we provide references or instructions for obtaining each one. All theoretical results are stated with explicit assumptions and are accompanied by complete proofs in Appendix 4.

**Acknowledgements.**  M. Seleznova acknowledges travel funding from the LMU Postdoc Support Fund for presenting this work at ICLR. G. Kutyniok acknowledges support from the gAIn project, which is funded by the Bavarian Ministry of Science and the Arts (StMWK Bayern) and the Saxon Ministry for Science, Culture and Tourism (SMWK Sachsen).

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

# Supplementary Material

This supplementary material is organized as follows:

- **Section A** provides a detailed discussion of related work, expanding on the brief overview given in the main paper.
- **Section B** presents additional theoretical results and complete proofs of the main propositions and theorems introduced in the paper.
- **Section C** describes the implementation details of GradPCA algorithm and the baseline methods used in our experiments.
- **Section D** discusses the computational cost considerations, including a breakdown of the components involved, empirical runtime comparisons, and an estimate of total compute usage.
- **Section E** reports additional experimental results, including CIFAR-10 benchmark and a set of ablation studies for GradPCA variants.

**Use of LLMs.** Large language models (LLMs) were used to assist with editing and polishing the writing of this paper. All substantive contributions—including theoretical results, empirical findings, and experimental design—were developed entirely by the authors without the aid of LLMs.

## A    RELATED WORKS

The field of out-of-distribution detection has witnessed rapid growth in recent years, with over 15,000 publications since 2021 referencing either *"OOD detection"* and *"out-of-distribution detection"*. This surge in research activity has made it increasingly challenging to maintain a comprehensive view of methodological advances and theoretical developments. Still, this section complements the overview provided in Section 2 by offering a broader survey of related work. We organize existing OOD detection methods according to our proposed taxonomy of *regularity-based* and *abnormality-based* approaches, which provides a unified lens for understanding a wide range of techniques in the literature.

### A.1    SURVEYS AND BACKGROUND ON OOD DETECTION

Out-of-distribution detection has become a central topic in machine learning, particularly in the context of safety and robustness. For a broad overview of the field, Tran (2023) maintain an extensive and regularly updated repository of OOD detection research. The foundational work by Hendrycks & Gimpel (2017) introduced one of the first widely adopted baselines for OOD detection. More recent surveys, such as those by Yang et al. (2021) and Salehi et al. (2022), provide systematic categorizations of the growing variety of techniques developed in this rapidly evolving area.

### A.2    REGULARITY-BASED OOD DETECTION

Many OOD detection methods rely on the structured nature of in-distribution (ID) data in a given feature space. These *regularity-based* methods model the geometry or statistics of ID samples—using representations, decision boundaries, or statistical criteria—to flag inputs that deviate from this structure. Due to the nature of this class, regularity-based methods usually require a *training stage* to learn the properties of the ID data. While Section 2 provides a unified overview, here we expand on specific approaches.

**Proximity-based approaches.** A wide range of methods rely on the distance of test points to a certain learned regular region in the feature space. The Mahalanobis detector (Lee et al., 2018) fits class-conditional Gaussians and scores inputs by their distance to the closest class mean. KNN-based detectors (Sun et al., 2022b) operate in the penultimate-layer space, identifying OOD inputs by their distance to nearest neighbors on the ID feature manifold.

*Prototype-based detectors* (van Amersfoort et al., 2020; Du et al., 2022) compute per-class feature centers—prototype vectors, similar to the class means used in GradPCA or Mahalanobis—and

measure similarity between a test input and the nearest prototype. Since these methods often include additional regularization (e.g., energy shaping or uncertainty constraints) to improve ID–OOD separation, they can be viewed as hybrids that combine regularity-based modeling with abnormality-oriented scoring techniques.

Gram-matrix consistency (Sastry & Oore, 2020) and likelihood-ratio scoring (Ren et al., 2019) offer alternative strategies for exploiting structural regularities in ID data. In particular, the relationship between the activation Gram matrix and OOD detection explored in (Sastry & Oore, 2020) resonates with our theoretical discussion about the covariance matrix, which provides principled guarantees for spectral detection.

**Spectral approaches.** Spectral methods constitute a key class of regularity-based OOD detectors. A canonical example is kernel PCA, introduced by Schölkopf et al. (1998) and later applied to novelty detection Hoffmann (2007), who measured novelty via the squared distance to a learned principal subspace in feature space.

More recently, Guan et al. (2023) revisited PCA-based OOD detection, identifying the limitations of conventional PCA and proposing a regularized reconstruction error, further enhanced by combining it with energy-based scoring. Building on this and (Sun et al., 2022b), Fang et al. (2024b) introduced a kernel PCA detector using generic nonlinear kernels (e.g., RBF, plynomial) to improve separation between ID and OOD inputs within the principal component subspace.

Other recent methods apply spectral principles in novel, non-covariance-based ways. Li et al. (2024) proposed SeTAR, a training-free, post-hoc detector that applies spectral analysis directly to pretrained model weights, arguing that the low-rank directions preserve ID-relevant structure while suppressing OOD-sensitive responses. Wu et al. (2024b) took a complementary approach, enforcing neural collapse during training to align ID features with a principal subspace and constrain OOD features to lie in the orthogonal complement. This hybrid method combines regularity modeling with supervised feature separation.

**Gradient-based approaches.** While the majority of gradient-based OOD detectors align more closely with the abnormality-based category, a few recent methods employ spectral regularity explicitly in the gradient space. In particular, GradOrth (Behpour et al., 2023) operates in loss-gradient space and applies SVD to last-layer activations to estimate principal directions and tranfer them to the gradient space. In contrast, Wu et al. (2024a) project input gradients onto class-mean directions, avoiding SVD altogether for efficiency.

Although these methods bear superficial similarity to GradPCA in that they use spectral reasoning and gradient space representations, they lack the theoretical grounding of our approach. In GradOrth, it is unclear why the SVD of activation covariances should align with meaningful directions in the gradient space—there is no formal justification for this connection. In contrast, GradPCA performs SVD on the empirical NTK, which is the true dual of the gradient covariance matrix and thus guaranteed to share its spectrum. Moreover, GradOrth's reliance on large batches to estimate this structure limits its scalability to settings with many classes.

The approach of Wu et al. (2024a), while more scalable, makes a strong assumption that class-mean gradients themselves represent principal directions. However, these means are generally neither orthogonal nor aligned with the dominant eigenvectors of the gradient covariance. GradPCA addresses this by using class means as a compressed basis to approximate the NTK's leading eigenspace, rather than assuming they are eigenvectors themselves.

Another regularity-based method leveraging gradients was introduced by Sun et al. (2022a), who use class-conditional gradient distributions and a Mahalanobis-distance score for OOD detection. A related recent approach is GROOD (ElAraby et al., 2025), which computes gradients with respect to a synthetic OOD prototype and uses nearest-neighbor distances in gradient space as the detection score. In contrast to GradPCA, these methods do not use spectral decomposition and thus do not fall3 within the spectral framework developed in this paper.

### A.3 ABNORMALITY-BASED OOD DETECTION

A complementary line of research focuses on detecting OOD inputs by monitoring deviations in model behavior, rather than modeling the structure of ID data. As discussed in Section 2, we refer to these as abnormality-based methods. They evaluate model responses to unfamiliar inputs by tracking indicators such as uncertainty, confidence degradation, or anomalous activation patterns. Unlike regularity-based approaches, which characterize the ID data structure in a well-defined feature space, these methods rely on behavioral cues that arise when the model encounters OOD samples—giving rise to a broad range of detection strategies.

**Confidence-based approaches.** Many abnormality-based OOD detection methods define scoring functions on the model's output logits to estimate prediction confidence. These scores—such as softmax maximum, energy, or temperature-scaled outputs—do not correspond to distances in any well-defined feature space, as they lack a reference point or metric structure. Moreover, these methods are post-hoc: they do not estimate or rely on the structure of ID data, but rather assume that OOD samples induce abnormal confidence patterns. For these reasons, confidence-based methods are best categorized as abnormality-based.

A foundational method is Maximum Softmax Probability (MSP) (Hendrycks & Gimpel, 2017), which uses the model's top predicted class probability as an OOD score. ODIN (Liang et al., 2018) improves upon MSP by applying temperature scaling and gradient-based input perturbations to enhance confidence separation between ID and OOD samples. Energy-based detection (Liu et al., 2020) generalizes these ideas by scoring inputs based on the log-sum-exp of logits.

Several methods further refine confidence signals. ReAct (Sun et al., 2021) clips abnormally high activations in the penultimate layer to reduce overconfidence of OOD inputs, aiming to improve the scores like MSP or ODIN. DML (Zhang & Xiang, 2023) decouples the max-logit score into cosine similarity and norm components, learning a weighted combination to better separate ID and OOD examples. Finally, DICE (Sun & Li, 2022) identifies a sparse, high-confidence subnetwork by pruning low-magnitude weights, aiming to sharpen the model's output distribution and improve OOD discrimination.

**Gradient-based approaches.** A growing number of abnormality-based methods leverage gradient information to detect OOD samples by assuming that such inputs trigger distinct or unstable model responses. These approaches do not model in-distribution structure explicitly but instead define OOD based on deviations in gradient norms, patterns, or attribution behavior.

Lee & AlRegib (2020) introduced the *feature-extraction hypothesis*, proposing that the norm of the loss gradient can indicate how well the model understands a given input. This and a follow up work (Lee et al., 2022) showed that OOD or corrupted samples often yield larger or more erratic gradients, making gradient magnitude a useful proxy for abnormality. GradNorm (Huang et al., 2021) builds on this idea by computing the $\ell_1$-norm of gradients from the KL divergence between the model output and a uniform distribution, observing that ID inputs typically produce stronger gradient signals. Kwon et al. (2020) proposed a Fisher score for anomaly detection in autoencoders, interpreting reconstruction gradients as features and arguing that anomalous inputs require more complex updates.

Other methods analyze gradient-derived attribution maps. In particular, GAIA (Chen et al., 2023) detects OOD inputs by analyzing disruptions in gradient-based attribution maps such as saliency and integrated gradients. These are *input gradients*—computed with respect to the input, not model parameters as in GradPCA. GAIA method is post-hoc and model-agnostic, relying on irregularities in explanation behavior rather than ID modeling.

Some approaches incorporate gradient behavior during training. Sharifi et al. (Sharifi et al., 2024) introduce a regularization scheme to enforce locally consistent OOD scores, combining it with an energy-based sampling strategy to expose the model to diverse outlier patterns.

### A.4 NEURAL TANGENT KERNEL ALIGNMENT

GradPCA is inspired by insights from the Neural Tangent Kernel (NTK) framework, originally introduced by Jacot et al. (Jacot et al., 2018). While classical NTK theory considers the infinite-width limit where the kernel remains fixed during training, recent work has shifted toward studying the

empirical NTK and its dynamic alignment to the target function—termed *NTK alignment* (Baratin et al., 2021; Shan & Bordelon, 2021; Atanasov et al., 2022).

NTK alignment has been linked to improved generalization (Chen et al., 2020) and builds on earlier work in kernel methods, where alignment between kernel and target label structures was formally studied (Cristianini et al., 2001; Gönen & Alpaydın, 2011). Empirical studies have shown that this alignment emerges early in training (Fort et al., 2020), with recent works identifying multi-phase alignment dynamics (Shan & Bordelon, 2021; Atanasov et al., 2022), implicit regularization effects (Baratin et al., 2021), and block-structured NTKs aligned with class semantics (Seleznova & Kutyniok, 2022; Seleznova et al., 2023).

For GradPCA, this block structure is particularly important: it enables an efficient approximation of the NTK using only class-mean gradients. This insight underpins the core design of GradPCA, linking NTK alignment theory with a practical OOD detection mechanism.

# B    ADDITIONAL THEORY AND PROOFS

## B.1    SUPPORT-BASED OOD DETECTION

In this section, we provide further motivation for the support-based formulation of the OOD detection problem introduced in Section 2.

**No free lunch for OOD detection.**    Most of the OOD detection literature adopts a probabilistic view, where a detector $D : \mathcal{X} \to \{0, 1\}$ must decide whether a sample $x \in \mathcal{X}$ is *more likely* to be drawn from the in-distribution $\mu_{\text{id}}$ or some out-distribution $\mu_{\text{ood}}$. However, $\mu_{\text{ood}}$ is unknown in practice. Typically, we only have access to samples from $\mu_{\text{id}}$, and aim to construct a detector that performs well across a range of possible, unseen $\mu_{\text{ood}}$ distributions (e.g., different OOD datasets).

This objective is fundamentally ill-posed: without assumptions on $\mu_{\text{ood}}$, no detector can guarantee reliable performance. The following result formalizes this observation.

**Theorem B.1.** *(No free lunch) For any OOD detector $D$ and fixed in-distribution $\mu_{\text{id}}$, there exists an out-distribution $\mu_{\text{ood}}$ such that the performance of $D$ is no better than random guessing.*

*Proof.* Let $D : \mathcal{X} \to \{0, 1\}$ to be an OOD detector. Let $A_0 = \{x \in \mathcal{X} : D(x) = 0\}$ and $A_1 = \{x \in \mathcal{X} : D(x) = 1\}$. Denote $Y$ a random variable such that $Y = 1$ if $X \sim \mu_{\text{ood}}$ and $Y = 0$ if $X \sim \mu_{\text{id}}$. Then we have the following for the detector error probability:

$$\mathbb{E}[D(X) \neq Y] = P[A_1|Y = 0]P[Y = 0] + P[A_0|Y = 1]P[Y = 1]. \tag{11}$$

Moreover, noticing that $P[A_1|Y = 0] = \mu_{\text{id}}(A_1)$ and $P[A_0|Y = 1] = \mu_{\text{ood}}(A_0)$, and assuming that the samples are sampled from in- and -out-distribution with equal probability, i.e. $P[Y = 0] = P[Y = 1] = 1/2$, we can write:

$$\mathbb{E}[D(X) \neq Y] = \frac{1}{2}(\mu_{\text{id}}(A_1) + \mu_{\text{ood}}(A_0)) = \frac{1}{2}(1 + \mu_{\text{id}}(A_1) - \mu_{\text{ood}}(A_1)). \tag{12}$$

If $\mu_{\text{id}}$ and $D$ are fixed but $\mu_{\text{ood}}$ is arbitrary, one can always construct a distribution $\mu_{\text{ood}}$ such that $\mu_{\text{ood}}(A_1) \leq \mu_{\text{id}}(A_1)$, causing the detector to perform no better than random guessing.    □

**Likelihood vs. support.**    In the probabilistic view of OOD detection, the optimal detector for a given pair $(\mu_{\text{id}}, \mu_{\text{ood}})$ naturally depends on likelihoods under both distributions. Specifically, as follows from the proof of Theorem B.1, the set of inputs flagged as OOD by the ideal detector should satisfy:

$$A_1^* \in \arg\min_{A_1} \mu_{\text{id}}(A_1) - \mu_{\text{ood}}(A_1), \tag{13}$$

which yields the likelihood-based optimal detector:

$$D^*(x) := \begin{cases} 1, & \mu_{\text{id}}(x) < \mu_{\text{ood}}(x) \\ 0, & \mu_{\text{id}}(x) \geq \mu_{\text{ood}}(x). \end{cases} \tag{14}$$

However, this formulation leads to a conceptual issue: the classification of a fixed input $x \in \mathcal{X}$ as ID or OOD may vary depending on the choice of $\mu_{\text{ood}}$, even though $\mu_{\text{id}}(x)$ remains constant. This

is counterintuitive in practical OOD detection, where the goal is to determine whether a model can reliably predict the label of $x$. Since the choice of $\mu_{\text{ood}}$ does not influence the model's prediction for $x$, it should not determine whether $x$ is considered OOD.

In contrast, the support-based perspective avoids this issue by providing an optimal detector that is independent of $\mu_{\text{ood}}$. Here, we define $\mu_{\text{id}}$ as the distribution over inputs the model can correctly classify, and any input outside $\text{supp}(\mu_{\text{id}})$ is considered beyond the model's competence. This aligns better with the practical objective of OOD detection: identifying inputs where the model is likely to fail.

## B.2 Sufficient Condition for Spectral OOD Detection (Theorem 4.1)

**Theorem** (Sufficient condition for spectral OOD detection). *Let $X \sim \mu_{\text{id}}$ and $h : \mathcal{X} \to \mathbb{R}^P$ be any function from $L^2(\mu_{\text{id}})$. Consider the covariance matrix*

$$\mathbf{S}(h) := \mathbb{E}[h(X)h(X)^\top]. \tag{15}$$

*Let $\mathcal{P}h(x)$ be the orthogonal projection of $h(x)$ onto the range of $\mathbf{S}(h)$. For any $x \in \mathcal{X}$, if $\|\mathcal{P}h(x)\|_2 < \|h(x)\|_2$ and $h$ is continuous at $x$, then $x$ is OOD.*

*Proof.* Since the row space of a matrix is perpendicular to its kernel, $\boldsymbol{v} := h(x) - \mathcal{P}h(x)$ is in the kernel of $\mathbf{S}(h)$, i.e. $\mathbf{S}(h)\boldsymbol{v} = 0$. Hence

$$\|\boldsymbol{v}^\top h\|_{L^2(\mu_{\text{id}})}^2 = \boldsymbol{v}^\top \left[ \int h(x')h(x')^\top d\mu_{\text{id}}(x') \right] \boldsymbol{v} = \boldsymbol{v}^\top \mathbf{S}(h)\boldsymbol{v} = 0. \tag{16}$$

Now we will prove the result by contradiction. Suppose $x \in \text{supp}(\mu_{\text{id}})$. Note that since

$$\boldsymbol{v}^\top h(x) = \|h(x)\|_2^2 - \|\mathcal{P}h(x)\|_2^2 > 0, \tag{17}$$

by the continuity of $h$ at $x$, there exists some ball $B_\epsilon(x)$ such that $\boldsymbol{v}^\top h(x') \geq \frac{1}{2}\boldsymbol{v}^\top h(x)$ on $B$. Since $x \in \text{supp}(\mu_{\text{id}})$, $\mu_{\text{id}}(B_\epsilon(x)) > 0$. Thus we have

$$\|\boldsymbol{v}^\top h\|_{L^2(\mu_{\text{id}})}^2 \geq \int_{x' \in B_\epsilon(x)} (\boldsymbol{v}^\top h(x'))^2 d\mu_{\text{id}}(x') \geq \frac{1}{4}\mu_{\text{id}}(B_\epsilon(x)) \cdot (\boldsymbol{v}^\top h(x))^2 > 0. \tag{18}$$

However, this contradicts equation 16. Thus $x \notin \text{supp}(\mu_{\text{id}})$ and hence is OOD. $\square$

**Remark B.2.** Theorem 4.1 has the following properties:

1. **Distribution-free:** The result makes no assumptions on either the in-distribution $\mu_{\text{id}}$ or the out-distribution $\mu_{\text{ood}}$, making it completely distribution-free.

2. **Deterministic condition:** By adopting the support-based perspective, the theorem provides a deterministic condition that guarantees a point $x \in \mathcal{X}$ is out-of-distribution, rather than relying on high-probability bounds.

3. **Flexibility in feature map choice:** The theorem applies to any (almost everywhere) continuous function $h \in L^2(\mu_{\text{id}})$, which may represent logits, activations, gradients, or other feature maps useful for OOD detection. This generality allows the result to cover a wide range of existing spectral methods.

## B.3 Connection to PCA (Theorem 4.2)

Before proving Theorem 4.2, we restate a simplified version of the Davis–Kahan $\sin \Theta$ theorem (Davis & Kahan, 1970), which plays a central role in our analysis:

**Theorem B.3** (Davis–Kahan $\sin \Theta$ theorem). *Let $A$ and $B = A + E$ be self-adjoint matrices in $\mathbb{R}^{P \times P}$, with eigenvalues $\lambda_1 \geq \cdots \geq \lambda_P$ and $\hat{\lambda}_1 \geq \cdots \geq \hat{\lambda}_P$, respectively. Fix $k \in \{1, \ldots, P\}$, and let $V \in \mathbb{R}^{P \times k}$ and $\widehat{V} \in \mathbb{R}^{P \times k}$ contain the top-$k$ eigenvectors of $A$ and $B$, respectively. Define the spectral gap:*

$$\Delta := \inf |\lambda - \hat{\lambda}| : \lambda \in [\lambda_k, \lambda_1], , \hat{\lambda} \in (-\infty, \hat{\lambda}_{k+1}). \tag{19}$$

*Then the following bound holds:*

$$\left\|\sin\Theta\big(\mathrm{span}(V), \mathrm{span}(\widehat{V})\big)\right\|_2 \leq \frac{\|E\|_2}{\Delta}. \tag{20}$$

*In particular, for the corresponding orthogonal projections $\mathcal{P}_k = VV^\top$ and $\widehat{\mathcal{P}}_k = \widehat{V}\widehat{V}^\top$, we have:*

$$\|\widehat{\mathcal{P}}_k - \mathcal{P}_k\|_2 \leq \frac{2\,\|E\|_2}{\Delta}. \tag{21}$$

A generalization of this result is available in (Stewart & Sun, 1990); the version presented here follows the formulation in (Yu et al., 2015b). We now proceed to the proof of our main result.

**Theorem** (Robust OOD certificate for PCA). *Consider PCA applied to a matrix $\hat{\mathbf{S}} \in \mathbb{R}^{P \times P}$ (e.g., estimated from a noisy sample). Let $h$ and $\mathbf{S}(h)$ be as in Theorem 4.1, and assume the following:*

$$\|\mathbf{S}(h) - \hat{\mathbf{S}}\|_2 \leq \epsilon, \quad \mathrm{rank}(\mathbf{S}(h)) = C, \tag{22}$$

*i.e., $\hat{\mathbf{S}}$ approximates a rank-$C$ covariance matrix $\mathbf{S}(h)$. Let $\hat{\mathcal{P}}_C$ denote the orthogonal projector onto the top $C$ eigenvectors of $\hat{\mathbf{S}}$, and let $\lambda_C$ be the $C$-th largest eigenvalue of $\mathbf{S}(h)$. Then, for any input $x \in \mathcal{X}$, the following condition is sufficient to guarantee that $x$ is OOD:*

$$s_{\mathrm{PCA}}(x) < 1 - \frac{2\epsilon}{\lambda_C - \epsilon}, \quad s_{\mathrm{PCA}}(x) := \frac{\|\hat{\mathcal{P}}_C h(x)\|^2}{\|h(x)\|^2}. \tag{23}$$

*Proof.* We prove the result by applying the Davis-Kahan theorem to matrices $\mathbf{S}(h) \in \mathbb{R}^{P \times P}$ and $\hat{\mathbf{S}} \in \mathbb{R}^{P \times P}$. Suppose the eigenvalues of (symmetric positive definite) $\mathbf{S}(h)$ and $\hat{\mathbf{S}}$ are, respectively $\lambda_1 \leq \lambda_2 \leq \cdots \leq \lambda_P$ and $\hat{\lambda}_1 \leq \hat{\lambda}_2 \leq \cdots \leq \hat{\lambda}_P$. From the low-rank condition, we have that $\lambda_{C+1} = \cdots = \lambda_P = 0$.

To show the relationship between the score functions corresponding to $\mathbf{S}(h)$ and $\hat{\mathbf{S}}$, we notice the following:

$$|s(x) - s_{\mathrm{PCA}}(x)| = \left| \frac{\|\mathcal{P}h(x)\|_2^2 - \|\hat{\mathcal{P}}_C h(x)\|_2^2}{\|h(x)\|_2^2} \right| \leq \|\mathcal{P} - \hat{\mathcal{P}}_C\|_2, \tag{24}$$

where the inequality follows from the observation that projection matrices are symmetric and idempotent:

$$\mathcal{P} = \mathcal{P}^\top = \mathcal{P}^2. \tag{25}$$

Now we notice that the spectral gap defined in equation 19 is given by:

$$\Delta = \inf\{|\lambda - \hat{\lambda}| : \lambda \in [\lambda_C, \lambda_1], \hat{\lambda} \in (-\infty, \hat{\lambda}_{C+1})\} = \lambda_C - \hat{\lambda}_{C+1} \geq \lambda_C - \epsilon, \tag{26}$$

since $\lambda_{C+1}$ is zero, and $\|\mathbf{S}(h) - \hat{\mathbf{S}}\|_2 \leq \epsilon$ by assumption. Therefore, applying the Davis-Kahan theorem, we get:

$$\|\mathcal{P} - \hat{\mathcal{P}}_C\|_2 \leq \frac{2\epsilon}{\lambda_C - \epsilon}. \tag{27}$$

Together with equation 24, this completes the proof, this implies the following for the score functions:

$$|s(x) - s_{\mathrm{PCA}}(x)| \leq \frac{2\epsilon}{\lambda_C - \epsilon}. \tag{28}$$

Finally, from Theorem 4.1, we know that the following condition is sufficient to guarantee that $x \in \mathcal{X}$ is OOD (i.e., $x \notin \mathrm{supp}(\mu_{\mathrm{id}})$):

$$s(x) < 1, \quad s(x) = \frac{\|\mathcal{P}h(x)\|^2}{\|h(x)\|^2}, \tag{29}$$

where $\mathcal{P}$ is the projection on the range of $\mathbf{S}(h)$. Moreover, by equation 28, we have:

$$s(x) < s_{\mathrm{PCA}}(x) + \frac{2\epsilon}{\lambda_C - \epsilon}. \tag{30}$$

Therefore, the sufficient condition in terms of $s_{\mathrm{PCA}}(x)$ is given by

$$s_{\mathrm{PCA}}(x) + \frac{2\epsilon}{\lambda_C - \epsilon} < 1 \quad \Leftrightarrow \quad s_{\mathrm{PCA}}(x) < 1 - \frac{2\epsilon}{\lambda_C - \epsilon}. \tag{31}$$

$\square$

## B.4 SAMPLE COMPLEXITY OF COVARIANCE MATRIX

Before proving the main result, we first state the following concentration bound for empirical covariance estimation:

**Theorem B.4** (Rudelson–Vershynin (Rudelson & Vershynin, 2007, Thm. 3.1)). *Let $Y$ be a random vector in $\mathbb{R}^d$ satisfying $\|Y\|_2 \leq M$ almost surely and $\left\| \mathbb{E}[Y \otimes Y] \right\|_2 \leq 1$. For independent copies $Y_1, \ldots, Y_N$ set*

$$\Gamma_N := \frac{1}{N} \sum_{i=1}^N Y_i \otimes Y_i - \mathbb{E}[Y \otimes Y], \qquad a := C M \sqrt{\frac{\log N}{N}},$$

*where $C > 0$ is universal. Then*

*(i) **(Expectation)** If $a < 1$ one has $\mathbb{E} \|\Gamma_N\|_2 \leq a$.*

*(ii) **(Tail)** For every $t \in (0,1)$, $\Pr\{\|\Gamma_N\|_2 > t\} \leq 2\exp(-c\, t^2/a^2)$, where $c > 0$ is universal.*

Now we are ready to state and prove the main result of this section:

**Theorem B.5** (Sample complexity of covariance matrix). *Assume $\|h(x)\| \leq R$ for all $x \in \mathcal{X}$ and let*

$$\hat{\mathbf{S}}(h) := \frac{1}{N} \sum_{i=1}^N h(x_i) h(x_i)^\top \tag{32}$$

*be the approximation of $\mathbf{S}(h)$ from a sample of size $N$. Denote $s_{\mathrm{PCA}}$ and $\hat{s}_{\mathrm{PCA}}$ the PCA score functions for $\mathbf{S}(h)$ and $\hat{\mathbf{S}}(h)$ respectively (as defined in Eq. equation 9). Then for any $\delta \in (0,1)$ with probability at least $1 - \delta$ we have the following:*

$$|s_{\mathrm{PCA}}(x) - \hat{s}_{\mathrm{PCA}}(x)| \leq \frac{2\epsilon}{\Delta - \epsilon}, \quad \epsilon := \frac{2CR^2}{\Delta} \sqrt{\frac{\log(N/\delta)}{N}}, \tag{33}$$

*where $C > 0$ is a universal constant, and $\Delta := \lambda_C - \lambda_{C+1}$ is the spectral gap of $\mathbf{S}(h)$.*

*Proof.* We start by defining rescaled feature maps $\tilde{h}(x) = \frac{h(x)}{\sqrt{\lambda_1}}$, where $\lambda_1$ is the largest eigenvalue of $\mathbf{S}(h)$. This rescaling ensures $\|\mathbb{E}[\tilde{h} \otimes \tilde{h}]\|_2 = \frac{\|\mathbf{S}(h)\|_2}{\lambda_1} = 1$, as required by Theorem B.4. Additionally, we have $\|\tilde{h}(x)\|_2 \leq \frac{R}{\sqrt{\lambda_1}}$ almost surely. Then, applying Theorem B.4 with $M = \frac{R}{\sqrt{\lambda_1}}$, we obtain that for any parameter $t \in (0,1)$:

$$\Pr\left[\left\| \frac{1}{N} \sum_{i=1}^N \tilde{h}(x_i) \otimes \tilde{h}(x_i) - \mathbb{E}[\tilde{h} \otimes \tilde{h}] \right\|_2 > t\right] \leq 2\exp\left(-c\frac{t^2 N}{M^2}\right)$$

where $c > 0$ is a universal constant.

Setting $t = C' \frac{R}{\sqrt{\lambda_1}} \sqrt{\frac{\log(N/\delta)}{N}}$ for an appropriate constant $C'$, we get:

$$\Pr\left[\left\| \frac{1}{N} \sum_{i=1}^N \tilde{h}(x_i) \otimes \tilde{h}(x_i) - \mathbb{E}[\tilde{h} \otimes \tilde{h}] \right\|_2 > C' \frac{R}{\sqrt{\lambda_1}} \sqrt{\frac{\log(N/\delta)}{N}}\right] \leq \delta$$

Rescaling back to the original features, we have:

$$\|\hat{\mathbf{S}}(h) - \mathbf{S}(h)\|_2 = \lambda_1 \left\| \frac{1}{n} \sum_{i=1}^n \tilde{h}(x_i) \otimes \tilde{h}(x_i) - \mathbb{E}[\tilde{h} \otimes \tilde{h}] \right\|_2 \leq C' R \sqrt{\lambda_1} \sqrt{\frac{\log(n/\delta)}{n}}$$

Since $\lambda_1 \leq \|\mathbf{S}\|_2 \leq R^2$ (by our assumption that $\|h(x)\|_2 \leq R$), we can further simplify this to:

$$\|\hat{\mathbf{S}}(h) - \mathbf{S}(h)\|_2 \leq CR^2 \sqrt{\frac{\log(N/\delta)}{N}}$$

for a universal constant $C$.

The remainder of the argument follows analogously to the proof of Theorem 4.2. □

## B.5 Necessary Condition for Spectral OOD Detection (Theorem 4.5)

**Theorem** (Necessary condition for spectral OOD detection). *Consider the same setting as in Theorem 4.1 and the spectral OOD detector $D_h$ defined in Eq. equation 7 with arbitrary threshold $\delta \in (0, 1]$. Then the following condition is necessary for the efficiency (i.e., non-zero sensitivity) of $D_h$:*

$$\text{rank}(\mathbf{S}(h)) < \dim(\{h(x) : x \in \mathcal{X}\}). \tag{34}$$

*Proof.* As per Theorem 4.1, the range of $\mathbf{S}(h)$ has to be included into the full image of the feature map $\text{span}(\{h(x) : x \in \mathcal{X}\}) \subset \mathbb{R}^P$, we have

$$\text{rank}(\mathbf{S}(h)) \leq \dim(\{h(x) : x \in \mathcal{X}\}). \tag{35}$$

In case of equality, we have $\|\mathcal{P}h(x)\| = \|h(x)\|$ for all $x \in \mathcal{X}$. Therefore, a detector of the form

$$D_h(x) := \mathbb{1}_{[0,\delta)}(s(x)), \quad s(x) := \frac{\|\mathcal{P}h(x)\|^2}{\|h(x)\|^2} \tag{36}$$

does not identify a single OOD point and, thus, has sensitivity (true positives rate) equal to zero. Therefore, by contradiction, strict inequality is necessary for effectiveness of $D_h$. $\qquad\square$

## C Implementation Details

In this section, we describe the implementation of GradPCA and all baselines, along with details about the pretrained models used in our experimental setup.

### C.1 GradPCA Implementation

We implement GradPCA as a custom class compatible with JAX transformations, leveraging the `struct` module from the Flax library (Heek et al., 2024). We chose JAX (Bradbury et al., 2018) (with Flax as the deep learning framework) for two main reasons: (1) its automatic differentiation system provides a clean, expressive interface for working with neural network gradients, which are central to our method; and (2) its just-in-time (JIT) compilation enables competitive runtime performance with minimal low-level optimization.

The GradPCA class includes several configurable parameters that control which variant of the method is executed:

- **Method:** Specifies how principal directions are computed. We support three options:
    1. *Block structure (default)*: Computes directions using class-mean gradients, as described in the main text.
    2. *Batch*: Computes gradients over a batch and estimates directions via SVD of the dual matrix.
    3. *GradOrth*: Implements the GradOrth method from recent literature (Behpour et al., 2023) (also included in the baselines) using SVD of the covariance matrix of the last-layer activations instead of the NTK for estimating principal components. for consistency with the original GradOrth paper, this variant uses loss gradients (with uniform vector as label) rather than output gradients.
- **Parameter subset:** Specifies which subset of model parameters to differentiate with respect to (as a list of keys in the parameter dictionary). In our main experiments, we use only the parameters from the last hidden layer. Additionally, we include ablations for CIFAR-10 (see Appendix E).
- **Spectral threshold $\epsilon$:** Used to truncate the spectrum of size $C$ at the smallest $K$ such that $\sum_{i=1}^{K} \lambda_i / \sum_{i=1}^{C} \lambda_i \geq \epsilon$. The default $\epsilon = 0.99$ typically retains the full or nearly full spectrum on moderate-class datasets such as CIFAR.
- **Aggregation:** A boolean flag controlling whether to aggregate gradients across all $C$ output heads (via summing) before PCA (when `True`), or to compute separate scores for each head (when `False`). While the non-aggregated version allows finer analysis, it is computationally demanding and practical only for small-class datasets like CIFAR-10. Our main benchmarks use the aggregated version.

- **Sparsification (DICE):** An option to sparsify gradients before PCA, roughly following the DICE method (Sun & Li, 2022). Sparsification is based on parameter magnitudes (not gradient values) and requires specifying a sparsity level $p$, which determines the fraction of parameters removed. The method applies sparsification independently to each parameter matrix.

The GradPCA variants discussed in the main text correspond to the following configurations: the default **GradPCA** uses the block structure method with $\epsilon = 0.99$, the parameter subset restricted to the last hidden layer (classifier) parameters, and aggregation enabled. **GradPCA-Batch** uses the batch method with $\epsilon = 0.97$. **GradPCA-Vec** disables aggregation. **GradPCA+DICE** enables the sparsification option with the default sparsity parameter $p = 0.8$. In the following paragraphs, we describe the high-level logic of the training and inference procedures for GradPCA.

**Training (offline) phase.** Training is initiated upon instantiating the GradPCA class and executed over the input dataset. At the end of training, the computed principal component vectors are stored as attributes of the class for use during inference. The procedure requires a training state (i.e., model and parameter values) and GradPCA configuration parameters. The input dataset must be provided as a function that returns a per-class data loader, allowing compatibility with varied loading pipelines. The training loop computes gradient class-means by sequentially evaluating gradients over batches of configurable size. This implementation is fully scalable to large datasets and has computational requirements comparable to a single training epoch. Once the gradient class-means are computed, the remaining steps follow directly from Algorithm 1.

**Inference (online) phase.** During inference, the principal components—stored as attributes of the GradPCA instance—are used to compute scores by projecting the input gradients onto these components. Since inference speed is critical for OOD detection, efficient parallelization plays a key role. The optimal strategy depends on factors such as batch size, the size of the parameter subset, the number of principal components, and hardware capabilities. In practice, it may be more efficient to parallelize over either the batch dimension or the principal components. Our implementation supports both options, though for consistency, we use parallelization over principal components in our runtime evaluations (Appendix D).

### C.2 BASELINES IMPLEMENTATION

For most baselines—excluding gradient-based methods GAIA and GradOrth—we use implementations from the Detectors library (Dadalto, 2023). This ensures consistent, efficient, and reproducible comparisons. To emphasize fairness in our main evaluations, we apply each baseline with the same hyperparameters across all benchmarks (matching our use of a single GradPCA variant without tuning). Below, we detail the specific settings used for each method.

**Max logits.** This baseline uses the maximum logit value as the OOD score and requires no additional parameters.

**ODIN.** We set the softmax temperature to 1000, the default in the Detectors library.

**Energy.** The softmax temperature is fixed at 1.0, following the library's default.

**DICE.** We set the sparsity parameter to $p = 0.7$, which specifies the fraction of weights removed from the last layer. This value is the default in Detectors and follows the recommendation from the original DICE paper (Sun & Li, 2022), particularly for ImageNet. As there is no principled strategy for selecting this parameter, we adopt the known optimal setting for ImageNet across all experiments. In most cases, this baseline performs reasonably well. However, we observe that it completely fails on the CIFAR-10 BiT-M model, resulting in an AUC of just 35%, while other logit-based methods succeed. This illustrates a failure mode where the core assumption of DICE—that pruning small weights at a given sparsity level improves ID-OOD separation—does not hold.

**Mahalanobis.** We extract features from the penultimate layer and estimate the class-conditional means and covariance matrix from samples (using mean aggregation and sample covariance, as in

Detectors). Since this method is computationally intensive, we use the full training data for CIFAR, but limit ImageNet training to a 50,000-sample subset. For the ImageNet benchmark (Table 2), we report average performance along with standard deviations over three independent runs.

**KNN.** We use penultimate-layer features and evaluate based on the 10 nearest neighbors. Because KNN inference time scales poorly with dataset size, we subsample 10,000 points for training in all settings to maintain acceptable runtime—even for CIFAR. report average performance along with standard deviations over three independent runs in the main benchmarks for this baseline.

**ReAct.** We apply ReAct to penultimate-layer features, using a clipping threshold of 0.9—the default value in Detectors. As with Mahalanobis, we use the full training split for CIFAR and a 50,000-sample subset for ImageNet. We report average performance along with standard deviations over three independent runs for ImageNet. Similar to DICE, this baseline performs well on several benchmarks but fails completely (AUC < 50%) on the CIFAR-10 BiT-M model (where DICE also fails) and the CIFAR-100 ResNet-34 model. This highlights the inconsistency of ReAct's core assumption—at least under a fixed hyperparameter setup—across our evaluation.

**GAIA.** GAIA is not included in the Detectors library, but the authors provide open-source code. While their implementation is architecture-dependent, they include support for ResNet-34 and BiT models, which enables direct comparison using their original setup.

**GradOrth.** GradOrth (Behpour et al., 2023) is a recent gradient-based spectral method that differs from GradPCA in two key ways: (1) it uses gradients of the loss function rather than the model output, and (2) it derives dominant directions via SVD on the last-layer activations, rather than NTK structure. Since no official implementation was released, we reimplemented GradOrth within the GradPCA framework based on the paper's description.

**Revisited PCA.** This baseline applies PCA to the penultimate-layer features and computes a regularized OOD score following Guan et al. (2023). We set the retained variance threshold to 0.9, as recommended in the original paper.

**Projected Gradients.** Following Wu et al. (2024a), this method projects gradients onto a low-dimensional subspace spanned by class-mean gradients. An auxiliary network—consisting of a BatchNorm layer followed by a single linear layer—is then trained on these projected gradients, and OOD detection is performed using the Energy score applied to this auxiliary network. As suggested in the original work, we train the auxiliary network for 3 epochs, using the full ID dataset for CIFAR benchmarks and 10% of the ID data for the ImageNet benchmark.

**Kernel PCA (CoRP).** CoRP applies Kernel PCA using a cosine–Gaussian kernel approximated through random Fourier features, with feature map $\phi(x) = \phi_{\mathrm{RFF}}(\phi_{\cos}(x))$, where the cosine normalization is given by $\phi_{\cos}(x) = x/\|x\|_2$ (Fang et al., 2024b). This method involves several hyperparameters, including the number of random features $M$, the Gaussian kernel parameter $\gamma$, and the variance retention threshold $\epsilon$. Because performance is highly sensitive to $\gamma$, we evaluate $\gamma \in 0.5, 1.0, 3.0$ and select $\gamma = 0.5$, which yields the best results on ImageNet. Consistent with the original paper, we use $M = 4096$ and set $\epsilon = 0.9$.

**NCI.** The Neural Collapse Inspired (NCI) detector (Liu & Qin, 2025) computes a score by projecting the centered penultimate layer feature of an input onto the last-layer weight vector corresponding to the model's predicted class on this input. We use the basic NCI variant, which omits the additional filtering step based on the penultimate layer feature norm, and is therefore hyperparameter-free.

## C.3 MODELS

We use two classes of models in our experiments: ResNetV2 models from the Big Transfer (BiT) repository (Kolesnikov et al., 2020; Beyer et al., 2022), and ResNet34 models from the PyTorch Image Models (TIMM) library (Wightman, 2019). Since GradPCA is implemented in JAX and most baselines are implemented in PyTorch, our comparison setup relies on translating shared models between the two frameworks.

**ResNetV2 (BiT).** BiT models are publicly available and widely used in the OOD literature. These are ResNetV2 architectures pretrained either on ImageNet-1k (BiT-S) or on ImageNet-21k (BiT-M), with additional BiT-M variants fine-tuned for smaller datasets such as CIFAR (after pretraining). We use these released models in our experiments.

BiT models are supported in both JAX and PyTorch. We use the official implementations for both frameworks, enabling a direct comparison between GradPCA and baselines. We confirm that the Torch and JAX versions of these models yield nearly identical accuracy, ensuring consistent evaluation.

**ResNet34 (TIMM).** The TIMM library provides a wide range of pretrained models, including ones for smaller datasets like CIFAR, while many popular frameworks offer only ImageNet-pretrained versions. We select the ResNet34 model to align with the setup of the GAIA method (Chen et al., 2023), which reports strong results on this architecture. However, we note that their reported performance is based on a custom-trained version of ResNet34, which we cannot replicate using the standard TIMM model.

TIMM models are not directly available in JAX, so we implement a custom conversion. Despite matching the architecture exactly, direct translation leads to a noticeable accuracy drop—likely due to numerical inconsistencies and subtle differences in layer implementations between PyTorch and Flax. For instance, the converted model achieves 93% accuracy on CIFAR-10 (vs. 95% in PyTorch) and 72% on CIFAR-100 (vs. 78%).

To ensure fairness, we fine-tune the converted JAX models for 10 epochs using float32 precision and a small learning rate, correcting for conversion artifacts without altering the weights significantly. We then convert the fine-tuned model back to PyTorch, which restores the original performance. This strategy guarantees consistent model behavior across frameworks. We release all model checkpoints (both JAX and PyTorch) with our code.

## D    COMPUTATIONAL COSTS

In this section, we break down the computational performance of GradPCA and compare it with baseline methods. The total cost involves two key components: the *training (offline) phase* and the *inference (online) phase*. While online efficiency is critical for real-time deployment, the offline phase must also be scalable enough to run on standard hardware. We discuss each phase in turn below, and also review memory requirements and provide an estimate of the total compute time for the project.

**Inference (online) phase.** The inference-time performance of GradPCA and baselines is summarized in Table 3. GradPCA is highly efficient on CIFAR, evaluating over 2,000 samples per second—well within the range for real-time deployment. However, like all methods, its performance degrades on ImageNet, which presents significantly greater computational challenges.

| Methods | Training | CIFAR-10 | | CIFAR-100 | | ImageNet-1k | |
| --- | --- | --- | --- | --- | --- | --- | --- |
| | | ResNetV2-50 | ResNet34 | ResNetV2-50 | ResNet34 | ResNetV2-101 | ResNetV2-50 |
| Max logits | | 2828 s/s | 5049 s/s | 2572 s/s | 4347 s/s | 425 s/s | 427 s/s |
| MSP | | 2831 s/s | 5565 s/s | 2825 s/s | 5690 s/s | 459 s/s | 475 s/s |
| ODIN | | 2817 s/s | 5060 s/s | 2592 s/s | 5436 s/s | 456 s/s | 473 s/s |
| Energy | | 2815 s/s | 5750 s/s | 2648 s/s | 5123 s/s | 458 s/s | 476 s/s |
| DICE | | 2815 s/s | 5731 s/s | 2733 s/s | 5141 s/s | 457 s/s | 476 s/s |
| Mahalanobis | ✓ | 2538 s/s | 4915 s/s | 2108 s/s | 3881 s/s | 363 s/s | 376 s/s |
| KNN | ✓ | 449 s/s | 1462 s/s | 449 s/s | 1476 s/s | 88 s/s | 89 s/s |
| ReAct | ✓ | 2809 s/s | 5413 s/s | 2838 s/s | 5117 s/s | 459 s/s | 484 s/s |
| GAIA-A | | 139 s/s | 247 s/s | 129 s/s | 241 s/s | 303 s/s | 245 s/s |
| GAIA-Z | | 317 s/s | 489 s/s | 307 s/s | 472 s/s | 309 s/s | 390 s/s |
| **GradPCA** | ✓ | 2654 samples/s | 2440 samples/s | 1974 samples/s | 2311 samples/s | 103 samples/s | 128 samples/s |

Table 3: Runtime of OOD detection methods evaluated on NVIDIA RTX A6000. Each method allows batch-evaluation, the results in the table show the average runtime per sample when evaluating batches of size 128. The time includes standardized data loading routine (TFDS loader).

Our implementation parallelizes GradPCA over the principal components, making its runtime relatively insensitive to the number of classes or components, provided sufficient GPU memory is

available. The negligible difference in runtime between CIFAR-10 and CIFAR-100 supports this. The slowdown on ImageNet is primarily due to the increased cost of a single projection, resulting from the larger parameter matrix in the last hidden layer. This cost could potentially be reduced by further limiting the subset of parameters used in gradient computation.

**Training (offline) phase.** The offline phase of GradPCA involves computing average class Jacobians over an input dataset (or a subset thereof). Jacobians are evaluated batch-wise, with a configurable batch size. As a result, the training time scales approximately with the dataset size divided by the batch size and is comparable to the time required for a single training epoch.

In our experiments, using the ResNetV2-50 model and an NVIDIA RTX A6000 GPU, we observed the following training times over the full datasets: approximately 2 minutes for CIFAR-10, 10 minutes for CIFAR-100, and 90 minutes for ImageNet-1k.

**Memory requirements.** Like other regularity-based methods, GradPCA incurs a memory overhead for storing the principal component vectors. The number of components retained depends on the spectral threshold $\epsilon$, but we can estimate the *maximum* memory requirement as $O(CP)$, where $C$ is the number of classes and $P$ is the number of parameters per class-specific vector.

In our setup using the ResNetV2-50 model (float32 precision, accounting for the fact that $P$ scales with $C$ in the final layer), this corresponds to the following worst-case memory usage: approximately 7.8 MB for CIFAR-10, 78 MB for CIFAR-100, and 7.5 GB for ImageNet. While the worst-case cost on ImageNet is relatively high, the actual memory usage in most experiments is much lower. This is because the default $\epsilon = 0.99$ threshold typically retains fewer than half of the principal components.

**Total computational cost.** The main computational cost of our project comes from training on the ImageNet benchmarks, including ablation studies. Specifically, we conducted 15 ImageNet runs each for ResNetV2-50 and ResNetV2-101, totaling approximately 45 hours of training time on an NVIDIA RTX A6000 GPU, with under 5 hours needed for evaluation. In comparison, the CIFAR experiments required significantly less compute, amounting to fewer than 10 GPU hours in total. Overall, we estimate the total computational cost of our results to be under 60 GPU hours on an NVIDIA RTX A6000.

## E    ADDITIONAL EXPERIMENTS

This section presents additional experimental results not included in the main paper, specifically the CIFAR-10 benchmark and ablation studies on GradPCA variants.

### E.1    CIFAR-10 BENCHMARK

The setup of the CIFAR-10 benchmark is identical to that of CIFAR-100, as described in Section 5. As shown in Table 4, similar to the CIFAR-100 results, regularity-based methods achieve state-of-the-art performance on the pretrained model, while abnormality-based methods perform better on the non-pretrained model, with GAIA achieving the best results.

Due to the smaller number of classes, this benchmark is significantly more computationally accessible than the others. This makes it feasible to apply GradPCA-Vec and GradPCA-Batch, as well as to perform GradPCA on a much larger subset of parameters beyond just the penultimate layer—resulting in improved performance. While these variants may be impractical for benchmarks with a larger number of classes, we recommend their use on smaller benchmarks to fully leverage the capabilities of GradPCA.

### E.1.1    NEAR-OOD SETTING

To further assess GradPCA's robustness under more challenging conditions, we complement the experiments on standard OOD detection datasets in Table 4 with a near-OOD benchmark in Table 5, using CIFAR-10 as the ID dataset and CIFAR-100 as the OOD dataset. As expected, GradPCA's performance decreases slightly in this more difficult setting, similar to the behavior observed for the

**ResNetV2-50 (BiT-M)**

| Methods | SVHN FPR95↓ | AUROC↑ | Places FPR95↓ | AUROC↑ | LSUN-c FPR95↓ | AUROC↑ | LSUN-r FPR95↓ | AUROC↑ | iSUN FPR95↓ | AUROC↑ | Textures FPR95↓ | AUROC↑ | Average FPR95↓ | AUROC↑ |
|---|---|---|---|---|---|---|---|---|---|---|---|---|---|---|
| Max logits | 50.37 | 76.11 | 50.81 | 69.44 | 37.91 | 84.08 | 42.32 | 83.15 | 45.28 | 81.91 | 24.34 | 89.65 | 41.84 | 80.72 |
| MSP | 39.49 | 83.72 | 44.42 | 75.93 | 29.7 | 89.45 | 35.85 | 87.58 | 37.41 | 87.31 | 18.16 | 93.12 | 34.17 | 86.18 |
| ODIN | 50.35 | 76.14 | 50.82 | 69.44 | 37.92 | 84.09 | 42.29 | 83.16 | 45.28 | 81.92 | 24.33 | 89.66 | 41.83 | 80.74 |
| Energy | 60.77 | 72.3 | 55.52 | 67.12 | 46.99 | 81.06 | 49.87 | 80.69 | 53.08 | 79.06 | 30.42 | 87.53 | 49.44 | 77.96 |
| DICE | 99.67 | 40.58 | 100 | 13.14 | 99.74 | 43.25 | 99.88 | 44.5 | 99.88 | 44 | 100 | 22.87 | 99.86 | 34.72 |
| Mahalanobis | 9.07 | 98.19 | **1.64** | **99.6** | 6.06 | 98.65 | 17.36 | 97.04 | 16.24 | 97.13 | 0.02 | 99.97 | 8.4 | 98.43 |
| KNN | 1.99 | 99.42 | 5.52 | 98.74 | 2.15 | 99.43 | 9.77 | 98.11 | 9.49 | 98.03 | 0.18 | 99.94 | 4.85 | 98.95 |
| ReAct | 99.54 | 28.46 | 99.98 | 15.52 | 98.03 | 46.36 | 99.59 | 38.91 | 99.41 | 39.24 | 95.77 | 41.93 | 98.72 | 35.07 |
| GAIA-A | 41.53 | 90.09 | 18.16 | 96.51 | 24.36 | 94.67 | 35.86 | 92.51 | 35.88 | 92.18 | 10.74 | 97.87 | 27.75 | 93.97 |
| GAIA-Z | 57.6 | 87 | 89.31 | 68.1 | 51.87 | 82.62 | 97.72 | 48.22 | 92.07 | 57.25 | 9.13 | 98 | 66.28 | 73.53 |
| GradOrth | 58.25 | 85.49 | 97.89 | 59.96 | 81.39 | 80.69 | 78.20 | 80.29 | 77.50 | 81.16 | 50.50 | 90.83 | 73.95 | 79.74 |
| Revisited PCA | 29.06 | 90.29 | 37.90 | 82.26 | 24.36 | 93.75 | 23.08 | 94.16 | 17.39 | 95.19 | 15.91 | 94.58 | 24.95 | 91.04 |
| Proj. Grads | 35.28 | 90.00 | 36.52 | 88.36 | 21.48 | 95.09 | 23.61 | 94.57 | 24.13 | 94.78 | 12.52 | 96.66 | 25.59 | 93.24 |
| Kernel PCA (CoRP) | 6.17 | 98.70 | 13.29 | 97.31 | 5.07 | 98.91 | 55.99 | 89.45 | 58.07 | 87.93 | 0.12 | 99.96 | 23.12 | 95.38 |
| NCI | 23.15 | 94.25 | 16.14 | 94.35 | 10.26 | 97.32 | 14.10 | 96.71 | 15.94 | 96.95 | 2.27 | 99.04 | 13.64 | 96.44 |
| **GradPCA** | 6.92 | 98.67 | 23.16 | 94.41 | 15.1 | 97.21 | 36.76 | 92.97 | 37.65 | 92.74 | 1.33 | 99.73 | 20.15 | 95.96 |
| **GradPCA (block 4)** | 5.11 | 99.01 | 10.96 | 97.25 | 6.01 | 98.9 | 20.02 | 96.49 | 19.87 | 96.49 | 0.62 | 99.87 | 10.43 | 98 |
| **GradPCA-Vec** | 6.08 | 98.77 | 11.01 | 97.47 | 6.59 | 98.76 | 17.94 | 96.82 | 17.05 | 96.84 | 0.36 | 99.91 | 9.84 | 98.1 |
| **GradPCA+DICE** | 10.98 | 97.96 | 28.77 | 93.35 | 17.95 | 96.78 | 41.07 | 92.2 | 41.67 | 92.12 | 1.17 | 99.67 | 23.60 | 95.35 |
| **GradPCA-Batch** | **0.52** | **99.73** | 3.2 | 99.3 | 2.23 | 99.45 | 6.48 | 98.67 | 5.55 | 98.74 | 0.02 | 99.99 | 3.00 | 99.31 |

**ResNet-34 (TIMM)**

| Methods | SVHN FPR95↓ | AUROC↑ | Places FPR95↓ | AUROC↑ | LSUN-c FPR95↓ | AUROC↑ | LSUN-r FPR95↓ | AUROC↑ | iSUN FPR95↓ | AUROC↑ | Textures FPR95↓ | AUROC↑ | Average FPR95↓ | AUROC↑ |
|---|---|---|---|---|---|---|---|---|---|---|---|---|---|---|
| Max logits | 20.07 | 95.06 | 40.8 | 86.61 | 22.45 | 94.36 | 28.55 | 92.84 | 30.47 | 91.98 | 48.6 | 83.94 | 31.82 | 90.8 |
| MSP | 37.47 | 93.99 | 54.45 | 87.5 | 39.86 | 93.11 | 45.9 | 91.82 | 47.18 | 91.09 | 58.52 | 85.96 | 47.23 | 90.58 |
| ODIN | 20.07 | 95.06 | 40.78 | 86.61 | 22.45 | 94.36 | 28.52 | 92.84 | 30.47 | 91.98 | 48.6 | 83.94 | 31.81 | 90.8 |
| Energy | 19.15 | 95.16 | 40.02 | 86.68 | 21.57 | 94.46 | 27.68 | 92.94 | 29.51 | 92.08 | 47.75 | 83.95 | 30.95 | 90.88 |
| DICE | 16.01 | 95.38 | 44.33 | 82.84 | 20.52 | 93.83 | 30.96 | 90.62 | 32.09 | 89.81 | 52.31 | 79.5 | 32.7 | 88.66 |
| Mahalanobis | 69.9 | 87.63 | 56.02 | 89.2 | 59.66 | 90.4 | 45.49 | 92.2 | 47.78 | 91.83 | 51.6 | 91.31 | 55.08 | 90.43 |
| KNN | 37.25 | 94.3 | 42.12 | 92.22 | 31.81 | 94.93 | 29.99 | 95.33 | 31.97 | 94.82 | 42.03 | 92.77 | 35.86 | 94.06 |
| ReAct | 21.07 | 92.95 | 42.94 | 85.72 | 21.71 | 92.89 | 29.37 | 91.89 | 31.88 | 90.78 | 50.37 | 82.23 | 32.89 | 89.41 |
| GAIA-A | 25.07 | 95.12 | 29.54 | 94.44 | 13.45 | 97.53 | 18.16 | 96.9 | 19.95 | 96.66 | 16.76 | 96.79 | 20.49 | 96.24 |
| GAIA-Z | 4.39 | 98.79 | 32.34 | 91.7 | 5.27 | 98.87 | 13.06 | 97.32 | 12.56 | 97.47 | 9.02 | 98.21 | 12.77 | 97.06 |
| GradOrth | 52.35 | 88.18 | 37.60 | 91.13 | 34.75 | 92.57 | 29.71 | 94.05 | 32.60 | 93.10 | 53.50 | 89.80 | 40.42 | 91.97 |
| Revisited PCA | 24.08 | 94.85 | 35.81 | 90.83 | 20.80 | 94.95 | 27.26 | 94.15 | 32.69 | 93.29 | 45.35 | 87.51 | 30.59 | 92.6 |
| Proj. Grads | 33.21 | 90.10 | 47.73 | 82.76 | 30.16 | 90.04 | 41.66 | 86.85 | 44.21 | 85.34 | 58.63 | 74.32 | 42.6 | 84.9 |
| Kernel PCA (CoRP) | 46.63 | 93.19 | 41.70 | 92.51 | 36.72 | 94.27 | 33.54 | 94.81 | 36.9 | 94.19 | 43.43 | 93.15 | 39.15 | 93.68 |
| NCI | 23.65 | 95.22 | 24.56 | 94.26 | 20.51 | 95.69 | 28.21 | 95.25 | 13.04 | 96.41 | 20.46 | 94.64 | 21.74 | 95.25 |
| **GradPCA** | 48.56 | 90.27 | 34.75 | 92.29 | 31.29 | 93.81 | 27.09 | 95 | 30.02 | 94.19 | 49.22 | 91.54 | 36.82 | 92.85 |
| **GradPCA (block 3)** | 15.07 | 97.01 | 24.87 | 95.33 | 11.06 | 97.49 | 19.63 | 96.54 | 20.92 | 96.32 | 25.78 | 95.86 | 19.55 | 96.42 |
| **GradPCA-Vec** | 25.34 | 95.91 | 49.1 | 91.17 | 26.09 | 95.75 | 42.47 | 93.32 | 44.35 | 92.68 | 42.92 | 92.64 | 38.38 | 93.58 |
| **GradPCA+DICE** | 62.5 | 84.04 | 39.54 | 90.26 | 42.09 | 90.03 | 33.33 | 92.87 | 36.85 | 91.7 | 55.38 | 88.21 | 44.95 | 89.52 |
| **GradPCA-Batch** | 24.9 | 95.95 | 48.65 | 91.14 | 25.22 | 95.87 | 42.32 | 93.29 | 44.41 | 92.59 | 43.29 | 92.46 | 38.13 | 93.55 |

Table 4: **CIFAR-10.** For each architecture and evaluation metric (FPR95, AUROC), the best-performing method is shown in **bold**, and the second and third best methods are underlined.

baselines. However, no critical degradation is observed, confirming that this near-OOD scenario does not compromise the method.

| Model | Metric | MaxLogits | MSP | ODIN | Energy | DICE | Mahalanobis | KNN | ReAct | GAIA-A | GAIA-Z | GradPCA |
|---|---|---|---|---|---|---|---|---|---|---|---|---|
| BiT-M | FPR95 | 44.86 | 38.84 | 44.84 | 51.95 | 95.07 | 29.39 | 19.00 | 96.84 | 39.48 | 90.02 | 28.67 |
| | AUROC | 81.19 | 85.71 | 81.20 | 78.77 | 48.24 | 93.56 | 96.00 | 44.21 | 88.72 | 53.02 | 93.68 |
| TIMM | FPR95 | 53.73 | 62.71 | 53.77 | 52.82 | 55.71 | 68.19 | 54.04 | 57.87 | 55.12 | 51.78 | 53.86 |
| | AUROC | 81.43 | 83.90 | 81.43 | 81.47 | 77.78 | 85.37 | 89.36 | 76.08 | 86.92 | 84.63 | 86.32 |

Table 5: **CIFAR-10 as ID vs. CIFAR-100 as OOD.** Near-OOD detection benchmark. For each architecture and evaluation metric (FPR95, AUROC), the best-performing method is shown in **bold**, and the second and third best methods are underlined.

## E.2 ViT-B/16 ImageNet Benchmark

Since many recent OOD benchmarks include Vision Transformer (ViT) models Dosovitskiy et al. (2021), we have evaluated GradPCA on ViT-B/16 in addition to our main benchmarks. We used a publicly available checkpoint pretrained on ImageNet-21k and fine-tuned on ImageNet-1k from the Vision Transformer repository.

As shown below, GradPCA remains effective and competitive on this architecture. We note that prior work has observed that OOD detection performance is generally comparable between ResNetV2 (BiT) and ViT backbones (Zhang et al., 2023). Since GAIA is not directly applicable to ViT models and GradOrth does not provide a computationally feasible implementation (see Appendix C.2), we did not include these methods in the following table.

## E.3 Consistency with respect to random seed

In addition to the inconsistencies in OOD detection performance arising from architecture and training setup, discussed in the main text, recent work has shown that OOD performance can also vary substantially with changes to the random seed governing training stochasticity (Fang et al., 2024a). To assess the impact of this factor on GradPCA, we trained a VGG16 model on CIFAR-10 using ten different random seeds, yielding models with nearly identical accuracy but converging to different local minima. We then evaluated GradPCA on each of these models. The results, presented in Table 7, show that performance varies only mildly across seeds (within roughly a 3% AUC range).

| | Places | | SUN | | iNaturalist | | Textures | | Average | |
|---|---|---|---|---|---|---|---|---|---|---|
| **Methods** | FPR95 ↓ | AUROC ↑ | FPR95 ↓ | AUROC ↑ | FPR95 ↓ | AUROC ↑ | FPR95 ↓ | AUROC ↑ | FPR95 ↓ | AUROC ↑ |
| Max logits | 61.92 | 85.20 | 55.32 | 88.12 | 13.27 | 97.54 | 52.50 | 86.87 | 45.75 | 89.43 |
| MSP | 71.23 | 79.32 | 66.99 | 80.92 | 22.05 | 94.92 | 62.07 | 80.00 | 55.59 | 83.79 |
| ODIN | 61.92 | 85.20 | 55.32 | 88.12 | 13.27 | 97.54 | 52.49 | 86.87 | 45.75 | 89.43 |
| Energy | 57.60 | 85.99 | 49.73 | 89.18 | 11.29 | 97.90 | 47.87 | **87.80** | 41.62 | 90.22 |
| DICE | **48.66** | 87.82 | **41.81** | 90.69 | 15.20 | 96.73 | 60.20 | 86.96 | 41.47 | 90.55 |
| Mahalanobis | 67.60 | 81.76 | 61.12 | 85.68 | 4.75 | 99.09 | 54.30 | 86.62 | 46.94 | 88.29 |
| KNN | 69.02 | 88.23 | 62.08 | 86.06 | 12.48 | 97.63 | 50.30 | 87.57 | 48.47 | 88.37 |
| ReAct | 56.72 | 85.84 | 47.86 | 89.28 | 14.90 | 97.29 | 45.69 | 87.65 | 41.29 | 90.22 |
| **GradPCA** | 50.86 | **89.02** | 42.80 | **91.27** | **3.74** | **99.16** | 59.94 | 86.21 | **39.84** | **91.42** |

*(ViT-B/16)*

Table 6: **ViT-B/16 on ImageNet-1k.** For each architecture and evaluation metric (FPR95, AUROC), the best-performing method is shown in **bold**, and the second and third best methods are underlined.

No drastic fluctuations are observed, indicating that GradPCA exhibits stable and consistent behavior under changes in training randomness.

| | | SVHN | | Places | | LSUN-c | | LSUN-r | | iSUN | | Textures | | Average | |
|---|---|---|---|---|---|---|---|---|---|---|---|---|---|---|---|---|
| | | FPR95 | AUROC | FPR95 | AUROC | FPR95 | AUROC | FPR95 | AUROC | FPR95 | AUROC | FPR95 | AUROC | FPR95 | AUROC |
| Seeds | 1 | 57.535 | 89.273 | 56.612 | 87.827 | 48.508 | 92.049 | 49.038 | 91.285 | 52.287 | 90.543 | 65.518 | 87.962 | 54.920 | 89.820 |
| | 10 | 63.678 | 87.551 | 66.861 | 83.348 | 58.023 | 89.334 | 62.550 | 86.629 | 61.311 | 86.626 | 67.827 | 83.686 | 63.380 | 86.200 |
| | 42 | 46.405 | 92.246 | 62.643 | 85.251 | 43.950 | 92.244 | 55.298 | 88.811 | 56.352 | 88.824 | 65.501 | 87.054 | 55.020 | 89.070 |
| | 100 | 55.588 | 90.613 | 72.045 | 81.752 | 54.577 | 90.345 | 64.052 | 86.194 | 66.599 | 85.473 | 72.887 | 84.887 | 64.291 | 86.544 |
| | 1000 | 64.948 | 86.362 | 62.859 | 86.349 | 52.965 | 90.678 | 56.931 | 89.489 | 57.529 | 89.385 | 62.695 | 89.283 | 59.654 | 88.593 |
| | 5000 | 55.419 | 90.317 | 61.823 | 85.605 | 53.566 | 90.463 | 55.549 | 88.899 | 56.046 | 88.800 | 60.760 | 88.154 | 57.194 | 88.706 |
| | 10000 | 50.258 | 90.085 | 65.965 | 85.495 | 45.563 | 91.855 | 64.453 | 87.652 | 65.976 | 87.096 | 74.343 | 84.368 | 61.093 | 87.758 |
| | 20000 | 44.019 | 92.491 | 67.588 | 84.086 | 45.673 | 91.818 | 65.565 | 86.210 | 67.267 | 85.179 | 76.314 | 81.825 | 61.071 | 86.935 |
| | 42000 | 46.429 | 92.208 | 61.530 | 85.705 | 48.898 | 91.951 | 50.691 | 90.249 | 52.276 | 90.053 | 63.743 | 87.815 | 53.928 | 89.663 |
| | 100000 | 37.107 | 93.337 | 64.232 | 85.927 | 43.109 | 92.434 | 59.886 | 88.059 | 62.466 | 87.044 | 72.177 | 82.442 | 56.496 | 88.207 |

Table 7: **CIFAR-10.** Ablation over random seeds.

## E.4 ABLATION: PARAMETERS SUBSET SELECTION

A key property of GradPCA, which contributes to its generality, is that it can be applied to any subset of a network's parameters. Prior work on NTK alignment has shown that alignment strength varies considerably across layers (Lou et al., 2022; Baratin et al., 2021): early layers typically exhibit weaker alignment, whereas later layers show stronger alignment, with the maximum often occurring in intermediate layers. This suggests that applying GradPCA to intermediate layers may be particularly effective.

We therefore conduct ablations in which GradPCA is applied to different parameter subsets of the BiT-M and TIMM models on CIFAR10 and CIFAR100. Table 8 reports the results for CIFAR10, where GradPCA is applied separately to the parameters of each residual block ("blocks 1–4"). As shown, intermediate blocks indeed yield strong performance, often exceeding that of the default version. For BiT-M, the best results occur at block 4, while for TIMM they occur at block 3. The performance degrades substantially for the earlier layers (blocks 1–2), indicating that their gradients are less informative for OOD detection, likely due to weak alignment.

Results for CIFAR100 are presented in Table 9. To manage computational cost, each block is further divided into "units", each containing two convolutional layers. Similar to CIFAR10, the best performance is achieved using intermediate layers (block 3, unit 3 for BiT-M and block 4, unit 2 for TIMM).

Although the most informative parameters consistently reside in later blocks, the exact choice of the optimal subset does not transfer cleanly across models or datasets, complicating layer selection for new architectures. Our practical recommendation is therefore to rely on empirical evaluation when selecting parameter subsets. We also note that understanding how NTK alignment and gradient representations evolve across layers remains an open problem in deep learning theory, and progress in this direction may further inform optimal layer selection for GradPCA.

## E.5 ABLATION: TRAINING DATA FRACTION

Since GradPCA relies on approximating gradient class means using ID data, it is natural to ask how sensitive the method is to the amount of available training data, and whether computation can be reduced by using only a subset. To investigate this, we performed an ablation in which GradPCA was trained on varying fractions of the ImageNet training set using the BiT-S model, and we evaluated the

| | | SVHN | | Places | | LSUN-c | | LSUN-r | | iSUN | | Textures | | Average | |
|---|---|---|---|---|---|---|---|---|---|---|---|---|---|---|---|
| | | FPR95 | AUROC | FPR95 | AUROC | FPR95 | AUROC | FPR95 | AUROC | FPR95 | AUROC | FPR95 | AUROC | FPR95 | AUROC |
| BiT-M | GradPCA (block 4) | 5.11 | 99.01 | 10.96 | 97.25 | 6.01 | 98.90 | 20.02 | 96.49 | 19.87 | 96.49 | 0.62 | 99.87 | 10.43 | 98.00 |
| | GradPCA (block 3) | 8.09 | 98.49 | 21.54 | 95.73 | 8.57 | 98.45 | 60.11 | 88.81 | 55.57 | 89.09 | 0.36 | 99.93 | 25.71 | 95.08 |
| | GradPCA (block 2) | 58.98 | 87.00 | 5.30 | 98.83 | 47.76 | 91.16 | 88.12 | 72.10 | 85.08 | 73.59 | 0.75 | 99.69 | 47.66 | 87.06 |
| TIMM | GradPCA (block 4) | 44.46 | 92.41 | 54.99 | 90.58 | 56.20 | 91.29 | 58.58 | 92.02 | 57.31 | 91.75 | 64.40 | 90.41 | 55.66 | 91.41 |
| | GradPCA (block 3) | 15.07 | 97.01 | 24.87 | 95.33 | 11.06 | 97.49 | 19.63 | 96.54 | 20.92 | 96.32 | 25.78 | 95.86 | 19.55 | 96.42 |
| | GradPCA (block 2) | 99.41 | 37.63 | 96.80 | 54.71 | 97.16 | 42.67 | 97.08 | 59.20 | 95.86 | 62.37 | 86.15 | 72.87 | 95.41 | 54.41 |
| | GradPCA (block 1) | 99.62 | 21.78 | 95.10 | 51.93 | 96.84 | 41.35 | 96.68 | 47.89 | 95.86 | 51.52 | 93.34 | 50.61 | 96.74 | 44.18 |

Table 8: **CIFAR10.** Ablation of the parameter subset choice using residual blocks of ResNet50-V2 (BiT-M) and ResNet34 (TIMM).

| | | SVHN | | Places | | LSUN-c | | LSUN-r | | iSUN | | Textures | | Average | |
|---|---|---|---|---|---|---|---|---|---|---|---|---|---|---|---|
| | | FPR95 | AUROC | FPR95 | AUROC | FPR95 | AUROC | FPR95 | AUROC | FPR95 | AUROC | FPR95 | AUROC | FPR95 | AUROC |
| BiT-M | GradPCA (block 4, unit 3) | 31.76 | 94.70 | 50.41 | 86.83 | 46.52 | 92.83 | 65.32 | 86.29 | 70.89 | 82.27 | 9.68 | 97.61 | 45.76 | 90.09 |
| | GradPCA (block 4, unit 2) | 31.54 | 93.73 | 54.50 | 85.26 | 41.19 | 93.30 | 77.15 | 82.08 | 79.13 | 78.91 | 6.75 | 98.43 | 48.38 | 88.62 |
| | GradPCA (block 4, unit 1) | 24.62 | 95.38 | 52.63 | 88.10 | 42.37 | 93.09 | 86.22 | 79.17 | 87.14 | 75.27 | 4.67 | 98.89 | 49.61 | 88.32 |
| | GradPCA (block 3, unit 6) | 11.50 | 97.74 | 40.60 | 92.16 | 30.34 | 95.11 | 79.40 | 79.62 | 77.80 | 78.01 | 0.76 | 99.67 | 40.07 | 90.39 |
| | GradPCA (block 3, unit 5) | 20.27 | 96.48 | 29.75 | 93.50 | 25.79 | 95.23 | 77.21 | 82.35 | 77.15 | 79.69 | 1.26 | 99.57 | 38.57 | 91.14 |
| | GradPCA (block 3, unit 4) | 53.89 | 90.17 | 55.77 | 87.43 | 67.59 | 83.73 | 90.84 | 70.22 | 88.81 | 69.03 | 12.93 | 97.58 | 61.64 | 83.03 |
| | GradPCA (block 3, unit 3) | 12.07 | 97.80 | 4.11 | 99.07 | 13.43 | 97.44 | 27.68 | 95.30 | 34.57 | 93.79 | 0.05 | 99.89 | 15.32 | 97.21 |
| | GradPCA (block 3, unit 2) | 30.75 | 94.94 | 13.02 | 97.40 | 50.28 | 89.25 | 58.52 | 89.32 | 64.20 | 84.96 | 2.75 | 99.46 | 36.59 | 92.56 |
| | GradPCA (block 3, unit 1) | 5.95 | 98.61 | 0.07 | 99.98 | 35.20 | 90.98 | 23.57 | 96.06 | 28.50 | 94.63 | 0.02 | 100.00 | 15.55 | 96.71 |
| TIMM | GradPCA (block 4, unit 3) | 69.79 | 85.79 | 61.39 | 87.26 | 67.92 | 86.39 | 63.18 | 86.25 | 63.20 | 85.61 | 52.84 | 87.52 | 63.05 | 86.47 |
| | GradPCA (block 4, unit 2) | 68.63 | 85.32 | 62.75 | 87.96 | 57.61 | 87.47 | 65.01 | 87.50 | 63.46 | 87.34 | 53.57 | 89.23 | 61.84 | 87.47 |
| | GradPCA (block 4, unit 1) | 66.04 | 79.45 | 74.74 | 80.59 | 51.14 | 83.46 | 79.64 | 78.18 | 76.55 | 79.96 | 51.19 | 89.29 | 66.55 | 81.82 |
| | GradPCA (block 3, unit 6) | 86.05 | 64.90 | 89.76 | 62.09 | 78.25 | 69.12 | 92.19 | 62.30 | 87.76 | 68.23 | 47.75 | 87.67 | 80.29 | 69.05 |
| | GradPCA (block 3, unit 5) | 70.09 | 80.00 | 88.55 | 70.80 | 78.53 | 73.40 | 84.97 | 76.48 | 77.58 | 79.73 | 35.92 | 91.27 | 72.61 | 78.61 |
| | GradPCA (block 3, unit 4) | 84.47 | 68.94 | 92.25 | 59.93 | 87.79 | 62.75 | 93.44 | 57.56 | 91.27 | 62.18 | 53.69 | 83.25 | 83.82 | 65.77 |
| | GradPCA (block 3, unit 3) | 69.40 | 78.20 | 89.68 | 67.77 | 79.42 | 70.44 | 89.13 | 68.98 | 85.04 | 72.11 | 44.51 | 87.49 | 76.20 | 74.17 |
| | GradPCA (block 3, unit 2) | 53.84 | 86.16 | 89.97 | 66.72 | 70.04 | 74.26 | 88.23 | 73.64 | 84.69 | 74.91 | 39.79 | 89.70 | 71.09 | 77.57 |
| | GradPCA (block 3, unit 1) | 77.31 | 78.39 | 96.52 | 56.46 | 77.73 | 68.45 | 97.21 | 53.91 | 96.18 | 56.98 | 66.23 | 78.51 | 85.03 | 65.45 |

Table 9: **CIFAR-100.** Ablation of the parameter subset choice for GradPCA using individual units in blocks 3 and 4, for BiT-M ResNet50-V2 and ResNet34 (TIMM). All numbers rounded to two decimals.

resulting OOD performance. The results are shown in Table 10. Notably, training with as little as 10% of the ID data yields similar performance to that obtained with the full dataset, demonstrating that GradPCA is robust to this choice and that gradient class means can be reliably estimated even from relatively small data subsets.

| | | Places | | SUN | | iNaturalist | | Textures | | Average | |
|---|---|---|---|---|---|---|---|---|---|---|---|
| | | FPR95 | AUROC | FPR95 | AUROC | FPR95 | AUROC | FPR95 | AUROC | FPR95 | AUROC |
| BiT-S | 10% | 60.36 | 81.649 | 48.42 | 87.208 | 16.32 | 96.029 | 17.365 | 96.316 | 35.62 | 90.30 |
| | 20% | 61.93 | 81.327 | 50.44 | 86.853 | 17.80 | 95.731 | 17.827 | 96.254 | 37.00 | 90.04 |
| | 30% | 61.33 | 81.757 | 49.21 | 87.233 | 17.38 | 95.835 | 17.809 | 96.186 | 36.43 | 90.25 |
| | 40% | 61.05 | 81.781 | 48.78 | 87.345 | 17.63 | 95.758 | 17.738 | 96.211 | 36.80 | 90.27 |
| | 50% | 61.48 | 81.650 | 49.73 | 87.139 | 18.54 | 95.693 | 17.987 | 96.148 | 36.93 | 90.16 |
| | 60% | 60.90 | 81.738 | 49.03 | 87.142 | 17.84 | 95.742 | 17.489 | 96.261 | 36.32 | 90.22 |
| | 70% | 61.58 | 81.685 | 49.53 | 87.160 | 17.62 | 95.849 | 17.720 | 96.212 | 36.61 | 90.23 |
| | 80% | 61.63 | 81.761 | 49.51 | 87.208 | 18.11 | 95.758 | 18.111 | 96.159 | 36.84 | 90.22 |
| | 90% | 61.61 | 81.565 | 49.85 | 87.083 | 17.95 | 95.766 | 18.058 | 96.149 | 36.87 | 90.14 |

Table 10: **ImageNet (BiT-S).** Ablation of the training set fraction used to fit GradPCA.

### E.6 ABLATION: RETAINED VARIANCE THRESHOLD

In the default version of GradPCA, used throughout all benchmark tables, we set the spectral threshold parameter to $\epsilon = 0.99$. For datasets with a small number of classes—such as CIFAR-10—performance is largely insensitive to this parameter, as any $\epsilon > 0.90$ typically results in selecting all $C - 1$ principal components. However, for datasets with a large number of classes, the choice of $\epsilon$ can have a more significant effect. To assess its impact, we conduct ablation studies on ImageNet (Table 11) and CIFAR-100 (Table 12).

We draw two main observations from this ablation:

1. The optimal value of $\epsilon$ varies across datasets and model architectures.
2. GradPCA's performance is largely stable under reasonable changes to $\epsilon$.

While tuning $\epsilon$ may lead to small gains or losses (typically within 1–2% in detection accuracy), there is no clear strategy for selecting an optimal value in advance, as its impact appears to depend on the specific task. Based on these findings, we recommend choosing the largest $\epsilon$ that remains

|  | Methods | Places FPR95 | AUROC | SUN FPR95 | AUROC | iNaturalist FPR95 | AUROC | Textures FPR95 | AUROC | Average FPR95 | AUROC |
|---|---|---|---|---|---|---|---|---|---|---|---|
| BiT-M | GradPCA ($\epsilon = 0.90$) | 85.96 | 66.47 | 78.12 | 74.95 | 40.88 | 92.04 | 49.33 | 88.96 | 63.57 | 80.60 |
| | GradPCA ($\epsilon = 0.95$) | 83.16 | 69.62 | 74.67 | 77.65 | 38.14 | 92.79 | 51.69 | 88.67 | 61.91 | 82.18 |
| | GradPCA ($\epsilon = 0.97$) | 82.25 | 70.63 | 73.70 | 78.39 | 38.56 | 92.77 | 53.66 | 88.28 | 62.04 | 82.52 |
| | GradPCA ($\epsilon = 0.99$) | 83.12 | 71.55 | 75.16 | 78.68 | 45.84 | 91.69 | 60.74 | 86.49 | 66.22 | 82.10 |
| BiT-S | GradPCA ($\epsilon = 0.90$) | 72.39 | 77.45 | 61.88 | 83.74 | 32.52 | 92.73 | 19.28 | 95.92 | 46.52 | 87.46 |
| | GradPCA ($\epsilon = 0.95$) | 67.55 | 79.38 | 57.11 | 85.15 | 22.82 | 94.54 | 18.02 | 96.22 | 41.38 | 88.82 |
| | GradPCA ($\epsilon = 0.97$) | 66.08 | 80.12 | 55.28 | 85.66 | 20.98 | 95.09 | 18.38 | 96.25 | 40.18 | 89.28 |
| | GradPCA ($\epsilon = 0.99$) | 61.27 | 81.75 | 49.35 | 87.20 | 17.78 | 95.80 | 18.08 | 96.14 | 36.62 | 90.22 |

Table 11: **ImageNet-1k**: Ablation of the $\epsilon$ parameter.

|  | Methods | SVHN FPR95 | AUROC | Places FPR95 | AUROC | LSUN-c FPR95 | AUROC | LSUN-r FPR95 | AUROC | iSUN FPR95 | AUROC | Textures FPR95 | AUROC | Average FPR95 | AUROC |
|---|---|---|---|---|---|---|---|---|---|---|---|---|---|---|---|
| BiT-M | GradPCA ($\epsilon = 0.90$) | 21.12 | 95.38 | 31.20 | 93.04 | 12.04 | 97.76 | 58.40 | 86.73 | 61.57 | 85.97 | 4.01 | 99.09 | 31.39 | 93.00 |
| | GradPCA ($\epsilon = 0.95$) | 19.20 | 95.98 | 28.88 | 93.71 | 8.71 | 98.37 | 53.42 | 88.14 | 59.02 | 86.60 | 3.43 | 99.25 | 28.78 | 93.68 |
| | GradPCA ($\epsilon = 0.97$) | 17.84 | 96.38 | 28.90 | 93.72 | 8.95 | 98.34 | 52.69 | 88.47 | 57.91 | 86.98 | 3.28 | 99.26 | 28.26 | 93.86 |
| | GradPCA ($\epsilon = 0.99$) | 17.20 | 96.58 | 29.64 | 93.56 | 8.28 | 98.42 | 51.75 | 88.93 | 56.93 | 87.34 | 3.41 | 99.24 | 27.87 | 94.01 |
| TIMM | GradPCA ($\epsilon = 0.90$) | 61.15 | 90.13 | 55.73 | 88.11 | 66.17 | 85.93 | 64.94 | 84.95 | 66.85 | 83.96 | 66.09 | 86.08 | 63.49 | 86.53 |
| | GradPCA ($\epsilon = 0.95$) | 54.55 | 90.65 | 60.14 | 87.83 | 63.32 | 87.31 | 70.95 | 85.84 | 71.18 | 85.13 | 69.28 | 86.38 | 64.90 | 87.19 |
| | GradPCA ($\epsilon = 0.97$) | 56.25 | 90.47 | 61.97 | 87.36 | 61.43 | 87.93 | 73.67 | 84.18 | 72.86 | 83.76 | 70.76 | 85.03 | 66.16 | 86.46 |
| | GradPCA ($\epsilon = 0.99$) | 61.22 | 89.10 | 62.71 | 87.31 | 63.45 | 87.11 | 73.97 | 84.25 | 73.01 | 83.85 | 72.30 | 84.63 | 67.78 | 86.04 |

Table 12: **CIFAR-100**: Ablation of the $\epsilon$ parameter.

computationally practical, as this parameter directly controls the number of principal components retained—and therefore the memory cost of GradPCA (see the next section).

### E.7 REDUCING MEMORY COST

GradPCA is a spectral method, and its memory footprint is dominated by the number of retained eigenvectors. A principled way to reduce memory usage is to lower the retained variance threshold $\epsilon$, which controls how many top eigenvectors are kept.

Table 13 reports the number of retained eigenvectors ($N$) and the corresponding AUROC on the ImageNet BiT-M and BiT-S benchmarks, evaluated at different $\epsilon$ values. These results were extracted from the ablation in Appendix E.6.

Table 13: AUROC scores and number of retained eigenvectors ($N$) for GradPCA on ImageNet BiT-M and BiT-S, across varying retained variance thresholds $\epsilon$.

| ImageNet BiT-M | | | ImageNet BiT-S | | |
|---|---|---|---|---|---|
| $\epsilon$ | $N$ | AUROC | $\epsilon$ | $N$ | AUROC |
| 0.99 | 568 | 82.10 | 0.99 | 432 | 90.22 |
| 0.97 | 400 | 82.52 | 0.97 | 274 | 89.28 |
| 0.95 | 322 | 82.18 | 0.95 | 208 | 88.82 |
| 0.90 | 221 | 80.60 | 0.90 | 132 | 87.46 |

These results show that controlling $\epsilon$ allows up to a $5\times$ reduction in memory with only a modest drop in AUROC on ImageNet.

### E.8 ABLATION: GRADPCA+DICE SPARSITY PARAMETER

In this section, we present ablation results for the sparsity parameter $p$ used in the GradPCA+DICE variant of our method. GradPCA+DICE applies a mask to sparsify the gradient vectors, retaining only the top $(1 - p) \times 100\%$ of entries (by magnitude) within the selected parameter subset. Table 14 shows results on ImageNet-1k, and Table 12 reports results on CIFAR-100.

From Tables 15 and 14, we observe that the optimal choice of the sparsity parameter $p$ varies significantly across settings. At the same time, sparsification does not lead to substantial performance differences compared to the default GradPCA in most cases. We fix the default value to $p = 0.8$, as it provides the greatest benefit on the BiT-M ImageNet benchmark. This value is also close to the DICE-recommended setting of $p = 0.7$, which is considered optimal for ImageNet.

| | Methods | Places FPR95 | AUROC | SUN FPR95 | AUROC | iNaturalist FPR95 | AUROC | Textures FPR95 | AUROC | Average FPR95 | AUROC |
|---|---|---|---|---|---|---|---|---|---|---|---|
| ResNetV2-101 (BiT-M) | **GradPCA** | 83.12 | 71.55 | 75.16 | 78.68 | 45.84 | 91.69 | 60.74 | 86.49 | 66.22 | 82.10 |
| | **GradPCA** ($p=0.1$) | 82.71 | 71.70 | 74.58 | 78.90 | 44.46 | 91.88 | 58.82 | 86.92 | 65.14 | 82.35 |
| | **GradPCA** ($p=0.2$) | 82.17 | 71.84 | 73.63 | 79.20 | 42.30 | 92.22 | 56.04 | 87.68 | 63.54 | 82.74 |
| | **GradPCA** ($p=0.3$) | 81.22 | 72.19 | 72.03 | 79.69 | 39.42 | 92.62 | 52.56 | 88.43 | 61.31 | 83.23 |
| | **GradPCA** ($p=0.4$) | 80.49 | 72.72 | 70.66 | 80.31 | 37.81 | 92.87 | 49.79 | 89.01 | 59.69 | 83.73 |
| | **GradPCA** ($p=0.5$) | 79.32 | 73.20 | 69.08 | 80.84 | 36.54 | 93.03 | 47.59 | 89.46 | 58.13 | 84.13 |
| | **GradPCA** ($p=0.6$) | 78.75 | 73.83 | 68.16 | 81.38 | 36.05 | 93.25 | 45.69 | 89.88 | 57.16 | 84.58 |
| | **GradPCA** ($p=0.7$) | 77.55 | 74.67 | 65.90 | 82.22 | 36.49 | 93.21 | 44.87 | 90.10 | 56.20 | 85.05 |
| | **GradPCA** ($p=0.8$) | 76.97 | 75.51 | 64.82 | 82.95 | 37.68 | 93.10 | 44.07 | 90.40 | 55.88 | 85.49 |
| | **GradPCA** ($p=0.9$) | 76.38 | 76.41 | 63.81 | 83.57 | 39.55 | 92.87 | 43.22 | 90.73 | 55.74 | 85.90 |
| ResNetV2-50 (BiT-S) | **GradPCA** | 61.27 | 81.75 | 49.35 | 87.20 | 17.78 | 95.80 | 18.08 | 96.14 | 36.62 | 90.22 |
| | **GradPCA** ($p=0.1$) | 61.51 | 81.75 | 49.57 | 87.21 | 17.92 | 95.77 | 18.22 | 96.14 | 36.80 | 90.22 |
| | **GradPCA** ($p=0.2$) | 61.88 | 81.54 | 49.89 | 87.04 | 18.02 | 95.74 | 18.00 | 96.15 | 36.95 | 90.12 |
| | **GradPCA** ($p=0.3$) | 60.16 | 82.23 | 48.20 | 87.46 | 16.97 | 95.99 | 18.82 | 95.98 | 36.04 | 90.42 |
| | **GradPCA** ($p=0.4$) | 61.09 | 81.89 | 49.49 | 87.21 | 17.27 | 95.92 | 18.95 | 95.97 | 36.70 | 90.25 |
| | **GradPCA** ($p=0.5$) | 62.24 | 81.54 | 50.43 | 86.92 | 17.80 | 95.81 | 18.98 | 95.99 | 37.36 | 90.06 |
| | **GradPCA** ($p=0.6$) | 62.51 | 81.41 | 50.76 | 86.84 | 17.97 | 95.80 | 19.05 | 96.01 | 37.57 | 90.02 |
| | **GradPCA** ($p=0.7$) | 63.20 | 81.07 | 51.41 | 86.59 | 18.40 | 95.64 | 19.00 | 96.04 | 38.00 | 89.84 |
| | **GradPCA** ($p=0.8$) | 65.07 | 80.10 | 53.16 | 85.85 | 19.25 | 95.48 | 18.16 | 96.20 | 38.91 | 89.41 |
| | **GradPCA** ($p=0.9$) | 67.17 | 79.07 | 55.47 | 85.09 | 21.61 | 94.92 | 17.19 | 96.42 | 40.36 | 88.88 |

Table 14: **ImageNet-1k**: Ablation of the parameter $p$ in GradPCA+DICE.

| | Methods | SVHN FPR95 | AUROC | Places FPR95 | AUROC | LSUN-c FPR95 | AUROC | LSUN-r FPR95 | AUROC | iSUN FPR95 | AUROC | Textures FPR95 | AUROC | Average FPR95 | AUROC |
|---|---|---|---|---|---|---|---|---|---|---|---|---|---|---|---|
| ResNetV2-50 (BiT-M) | **GradPCA** | 17.2 | 96.58 | 29.64 | 93.56 | 8.28 | 98.42 | 51.75 | 88.93 | 56.93 | 87.34 | 3.41 | 99.24 | 27.87 | 94.01 |
| | **GradPCA+DICE** ($p=0.1$) | 17.67 | 96.31 | 29.58 | 93.39 | 9.50 | 98.18 | 48.69 | 89.98 | 55.29 | 87.90 | 3.85 | 99.09 | 27.43 | 94.14 |
| | **GradPCA+DICE** ($p=0.2$) | 17.52 | 96.43 | 29.06 | 93.54 | 10.26 | 98.07 | 49.37 | 90.13 | 55.79 | 88.05 | 3.98 | 99.11 | 27.66 | 94.22 |
| | **GradPCA+DICE** ($p=0.3$) | 16.78 | 96.59 | 27.71 | 93.91 | 9.31 | 98.23 | 48.73 | 90.17 | 54.72 | 88.27 | 3.52 | 99.18 | 26.79 | 94.39 |
| | **GradPCA+DICE** ($p=0.4$) | 17.09 | 96.62 | 27.62 | 94.03 | 8.92 | 98.34 | 48.66 | 90.00 | 54.64 | 88.23 | 3.30 | 99.24 | 26.71 | 94.41 |
| | **GradPCA+DICE** ($p=0.5$) | 17.78 | 96.57 | 28.77 | 93.85 | 8.65 | 98.40 | 50.70 | 89.44 | 56.25 | 87.85 | 3.53 | 99.22 | 27.61 | 94.22 |
| | **GradPCA+DICE** ($p=0.6$) | 17.85 | 96.55 | 29.02 | 93.79 | 8.44 | 98.42 | 51.23 | 89.19 | 56.40 | 87.72 | 3.50 | 99.22 | 27.74 | 94.15 |
| | **GradPCA+DICE** ($p=0.7$) | 17.88 | 96.53 | 29.46 | 93.64 | 8.18 | 98.46 | 51.76 | 88.80 | 56.67 | 87.41 | 3.46 | 99.22 | 27.90 | 94.01 |
| | **GradPCA+DICE** ($p=0.8$) | 18.11 | 96.57 | 30.77 | 93.34 | 8.12 | 98.46 | 55.72 | 87.43 | 59.53 | 86.27 | 3.59 | 99.20 | 29.31 | 93.54 |
| | **GradPCA+DICE** ($p=0.9$) | 16.72 | 96.98 | 32.33 | 93.00 | 7.19 | 98.66 | 62.57 | 85.02 | 64.50 | 84.31 | 3.16 | 99.34 | 31.08 | 92.88 |
| ResNet-34 (TIMM) | **GradPCA** | 61.22 | 89.10 | 62.71 | 87.31 | 63.45 | 87.11 | 73.97 | 84.25 | 73.01 | 83.85 | 72.30 | 84.63 | 67.78 | 86.04 |
| | **GradPCA+DICE** ($p=0.1$) | 59.39 | 89.48 | 63.04 | 87.08 | 61.03 | 87.90 | 75.75 | 83.04 | 74.09 | 82.75 | 72.80 | 83.78 | 67.68 | 85.67 |
| | **GradPCA+DICE** ($p=0.2$) | 56.02 | 90.24 | 61.99 | 87.31 | 58.14 | 88.66 | 75.21 | 82.89 | 73.60 | 82.68 | 71.52 | 83.93 | 66.08 | 85.95 |
| | **GradPCA+DICE** ($p=0.3$) | 55.25 | 90.44 | 60.95 | 87.34 | 56.99 | 88.84 | 74.40 | 82.84 | 73.03 | 82.60 | 70.88 | 83.89 | 65.25 | 85.99 |
| | **GradPCA+DICE** ($p=0.4$) | 55.39 | 90.49 | 61.68 | 87.20 | 57.09 | 88.89 | 75.37 | 82.58 | 73.60 | 82.35 | 71.34 | 83.66 | 65.75 | 85.86 |
| | **GradPCA+DICE** ($p=0.5$) | 55.36 | 90.30 | 61.60 | 87.13 | 58.36 | 88.42 | 74.76 | 82.68 | 73.23 | 82.68 | 71.45 | 83.77 | 65.79 | 85.88 |
| | **GradPCA+DICE** ($p=0.6$) | 58.87 | 89.75 | 62.40 | 86.96 | 61.99 | 87.62 | 74.89 | 83.20 | 73.85 | 82.73 | 72.53 | 83.58 | 67.42 | 85.64 |
| | **GradPCA+DICE** ($p=0.7$) | 61.06 | 89.33 | 62.96 | 87.17 | 63.43 | 87.17 | 74.43 | 84.24 | 73.29 | 83.74 | 73.40 | 84.30 | 68.05 | 85.99 |
| | **GradPCA+DICE** ($p=0.8$) | 61.21 | 89.10 | 61.27 | 87.84 | 64.81 | 86.62 | 70.59 | 86.40 | 69.95 | 85.79 | 70.24 | 86.10 | 66.34 | 86.98 |
| | **GradPCA+DICE** ($p=0.9$) | 64.90 | 87.98 | 63.27 | 87.06 | 68.20 | 85.60 | 73.69 | 85.35 | 72.94 | 84.75 | 73.40 | 84.98 | 69.40 | 85.95 |

Table 15: **CIFAR-100**: Ablation of the parameter $p$ in GradPCA+DICE.

## E.9 ABLATION: GRADPCA-BATCH BATCH SIZE

| | Methods | SVHN FPR95 | AUROC | Places FPR95 | AUROC | LSUN-c FPR95 | AUROC | LSUN-r FPR95 | AUROC | iSUN FPR95 | AUROC | Textures FPR95 | AUROC | Average FPR95 | AUROC |
|---|---|---|---|---|---|---|---|---|---|---|---|---|---|---|---|
| BiT-M | **GradPCA-Batch** ($N=1000$) | 0.75 | 99.62 | 4.00 | 99.02 | 3.27 | 99.22 | 9.43 | 98.07 | 8.13 | 98.23 | 0.02 | 99.99 | 4.27 | 99.02 |
| | **GradPCA-Batch** ($N=2000$) | 0.64 | 99.70 | 3.65 | 99.17 | 2.68 | 99.36 | 8.21 | 98.35 | 6.88 | 98.49 | 0.02 | 99.99 | 3.68 | 99.18 |
| | **GradPCA-Batch** ($N=3000$) | 0.53 | 99.70 | 3.43 | 99.21 | 2.44 | 99.40 | 8.01 | 98.38 | 7.02 | 98.52 | 0.02 | 99.99 | 3.57 | 99.20 |
| | **GradPCA-Batch** ($N=4000$) | 0.56 | 99.73 | 3.31 | 99.27 | 2.26 | 99.45 | 6.83 | 98.61 | 5.80 | 98.72 | 0.02 | 99.99 | 3.13 | 99.29 |
| | **GradPCA-Batch** ($N=5000$) | 0.52 | 99.73 | 3.20 | 99.30 | 2.23 | 99.45 | 6.48 | 98.67 | 5.55 | 98.74 | 0.02 | 99.99 | 3.00 | 99.31 |
| TIMM | **GradPCA-Batch** ($N=1000$) | 24.47 | 96.04 | 51.05 | 90.83 | 26.13 | 95.79 | 46.08 | 92.85 | 48.30 | 92.12 | 47.02 | 91.81 | 40.51 | 93.24 |
| | **GradPCA-Batch** ($N=2000$) | 24.56 | 95.99 | 49.52 | 90.99 | 25.40 | 95.85 | 44.24 | 93.05 | 46.13 | 92.35 | 44.98 | 92.10 | 39.14 | 93.39 |
| | **GradPCA-Batch** ($N=3000$) | 24.43 | 95.99 | 48.57 | 91.10 | 25.11 | 95.87 | 42.16 | 93.28 | 44.29 | 92.58 | 42.72 | 92.47 | 37.88 | 93.55 |
| | **GradPCA-Batch** ($N=4000$) | 25.05 | 95.91 | 48.14 | 91.21 | 24.98 | 95.91 | 41.83 | 93.34 | 44.11 | 92.64 | 42.97 | 92.51 | 37.85 | 93.59 |
| | **GradPCA-Batch** ($N=5000$) | 24.90 | 95.95 | 48.65 | 91.14 | 25.22 | 95.87 | 42.32 | 93.29 | 44.41 | 92.59 | 43.29 | 92.46 | 38.13 | 93.55 |

Table 16: **CIFAR-10:** Ablation of batch size $N$ in GradPCA-Batch.

In this section, we present ablations for the batch size, denoted $N$, used in the GradPCA-Batch variant of our method. Computing principal components over a batch requires storing $N$ gradient vectors, which is not feasible for large parameter spaces or large batch sizes. In addition, performing SVD on the batch NTK matrix of size $N \times N$ scales poorly with $N$, further limiting practicality. As a result, we apply this variant only in the smaller CIFAR-10 benchmark, with results shown in Table 16. Nevertheless, where computationally feasible, GradPCA-Batch can provide additional performance gains.

## E.10 NTK BLOCK STRUCTURE FIGURES

In this section, we provide a full description of the NTK block-structure illustrations shown in Figure 1. Each heatmap visualizes the kernel values $\sum_{c=1}^{C} \Theta(x_i, x_j)$ computed over all network parameters. The kernels are evaluated on a random subset of the training data containing 12 samples per class, i.e., $x_i$ and $x_j$ range over 120 inputs in total. The samples are ordered so that examples

from the same class appear consecutively, i.e., $x_{12c}, \ldots, x_{12c+11}$ correspond to class $c \in \{1, \ldots, C\}$. Under this ordering, the diagonal blocks represent within-class interactions. All heatmaps are max-normalized, with white indicating the maximum kernel value and black indicating zero. The models included in the figure are: (1) ResNet20 trained on MNIST, (2) DenseNet40 trained on CIFAR10, and (3) VGG11 trained on CIFAR10. Each model was trained for 400 epochs with a batch size of 120 using SGD with Nesterov momentum 0.9 and weight decay $5 \cdot 10^{-4}$.

### E.11 GRADPCA UNDER LABEL NOISE

| Label noise | Test Acc | SVHN | | Places | | LSUN-c | | LSUN-r | | iSUN | | Textures | | Average | |
|---|---|---|---|---|---|---|---|---|---|---|---|---|---|---|---|
| | | FPR95 | AUROC | FPR95 | AUROC | FPR95 | AUROC | FPR95 | AUROC | FPR95 | AUROC | FPR95 | AUROC | FPR95 | AUROC |
| **0%** | 0.88 | 62.06 | 89.42 | 64.75 | 84.92 | 55.93 | 90.12 | 62.01 | 86.91 | 63.03 | 86.32 | 74.49 | 81.38 | 63.71 | 86.51 |
| **10%** | 0.80 | 69.75 | 83.21 | 73.60 | 79.83 | 59.43 | 86.02 | 69.76 | 82.84 | 71.05 | 82.25 | 77.98 | 78.11 | 70.26 | 82.04 |
| **20%** | 0.72 | 70.52 | 81.58 | 81.50 | 71.59 | 69.11 | 80.16 | 79.16 | 73.95 | 77.41 | 75.10 | 75.57 | 75.57 | 75.55 | 76.33 |
| **30%** | 0.63 | 82.30 | 74.48 | 85.43 | 68.79 | 83.45 | 72.01 | 83.85 | 70.33 | 82.41 | 70.80 | 80.90 | 68.65 | 83.06 | 70.84 |
| **40%** | 0.53 | 76.89 | 72.09 | 90.18 | 62.18 | 73.68 | 71.31 | 90.58 | 62.40 | 90.49 | 62.35 | 88.30 | 58.77 | 85.34 | 64.85 |
| **50%** | 0.41 | 90.46 | 59.08 | 91.95 | 58.55 | 90.95 | 61.74 | 91.83 | 59.30 | 91.44 | 59.55 | 82.96 | 62.26 | 89.98 | 60.41 |
| **60%** | 0.30 | 83.35 | 62.65 | 93.39 | 55.21 | 81.04 | 64.36 | 94.21 | 55.02 | 93.92 | 55.23 | 89.17 | 55.57 | 89.18 | 58.01 |
| **70%** | 0.21 | 88.72 | 57.42 | 94.33 | 51.52 | 75.26 | 64.69 | 94.02 | 51.73 | 94.20 | 51.60 | 90.45 | 52.86 | 89.49 | 54.30 |
| **80%** | 0.12 | 95.57 | 42.67 | 94.81 | 51.49 | 93.60 | 53.34 | 94.39 | 51.08 | 94.45 | 51.41 | 94.64 | 48.75 | 94.91 | 49.63 |

Table 17: **VGG16 on CIFAR-10 with label noise.** GradPCA OOD performance across noise levels.

In this section, we present experiments illustrating how label noise introduced during training affects the performance of GradPCA. Specifically, we train a VGG16 model on CIFAR-10 under ten levels of label noise, ranging from 0% to 90%. The label noise is fixed, i.e., labels are randomly corrupted once before training begins and remain unchanged throughout training. The training setup is as described in Appendix E.3. Results are reported in Table 17. We observe that GradPCA's performance degrades steadily as the noise level increases, mirroring the decline in test accuracy.

