# OpenReview forum: "GradPCA: Leveraging NTK Alignment for Reliable Out-of-Distribution Detection"
_ICLR.cc/2026/Conference — ICLR 2026 Poster_

### Official Review · Reviewer_zZXM · 2025-10-16

**Soundness:** 3
**Presentation:** 3
**Contribution:** 3
**Rating:** 6
**Confidence:** 5

**Summary:**

This work takes a Neural Tangent Kernel (NTK) perspective to investigate the Out-of-Distribution (OOD) detection problem. A novel detection method named GradPCA is proposed. In GradPCA, PCA is executed on the class-wise gradients of in-distribution (IND) data to obtain a low-dimensional subspace, and IND and OOD data are expected to exhibit well separability on this low-dimensional gradient subspace. Theoretical results are presented about the sufficient and necessary conditions for identifying a sample as OOD in spectral OOD detection, with discussions on how to select the feature mapping. Extensive empirical results validate the effectiveness of GradPCA. Besides, the relationships between OOD detection and some other issues such as consistency and feature quality are discussed.

**Strengths:**

1.	The NTK perspective and the associated GradPCA method are novel and beneficial to the OOD detection community.
2.	Discussions on the consistency and feature quality issues are appreciated and can provide new insights for OOD detection.
3.	The writing is good and easy to follow.
4.	Theories and extensive empirical results are provided.

**Weaknesses:**

**Major concerns**

1.	Unclear descriptions on the algorithm implementation

Some descriptions in Algorithm 1 are confusing, and thereby related details are suggested to be supplemented.

1.1	In line 4-6 in the offline training stage, what are the detailed executions and differences between line 4 and line 5? I guess that here PCs are obtained through eigendecomposition on the covariance matrix and its dual matrix, and please supplement detailed mathematical equations. But why there are two times of eigendecomposition? Then, given line 4 and line 5, which PCs are adopted to obtain the projection matrix in line 6? Please specify the calculations of the projection matrix clearly.

1.2	In line 9 in the online inference stage, I guess the PCA reconstruction error on gradients of a new sample is set as the detection score. But the definition for the projection matrix is missing, causing confusion on the detection score. Besides, why the norm of the projected gradients is normalized by that of the original gradients? Calculating reconstruction errors does not need the normalization of input norms. Please clarify this issue.

2.	Insufficient experiments.

While the experiments include several general detection methods, the comparisons lack more directly relevant, subspace-based baselines, as GradPCA itself is based on PCA and model gradients. The suggested baselines have been reviewed in the submission but are missing in empirical comparisons: (i) PCA [1] and Kernel PCA [2] applied to penultimate layer features, and (ii) the low-dimensional gradient subspace method explored in [3].

3.	Extended discussion on the consistency.

The inconsistency issue in OOD detection is rarely explored. The experiments in Figure 1 of this submission are executed across diverse benchmarks. Meanwhile, the inconsistency across multiple independently-trained modes (local optima) is highlighted in [4]. It would be appreciated to evaluate GradPCA on such independent modes to validate its consistency from the model side, which will further strengthen this work.

[1] Revisit pca-based technique for out-of-distribution detection. ICCV 2023.
[2] Kernel PCA for out-of-distribution detection. NeurIPS 2024.
[3] Low-Dimensional Gradient Helps Out-of-Distribution Detection. TPAMI 2024.
[4] Revisiting Deep Ensemble for Out-of-Distribution Detection: A Loss Landscape Perspective. IJCV 2024.

**Minor concerns**
4.	In the basic settings, the neural network function $f$ outputs a real number. Please specify the calculations of this outputted real number.
5.	The GradOrth method demonstrates quite poor detection performance across almost all datasets. It would be helpful for authors to clarify whether the hyper-parameters of GradOrth have been carefully tuned. If so, some explanations into the potential reasons for these results would be valuable.

**Questions:**

My questions are the five concerns listed above. I will raise my rating given all the concerns are well addressed.

---

> ### Author Response · Authors · 2025-11-20
>
> We thank the reviewer for their careful evaluation of our work and the valuable suggestions that helped us improve the quality of our paper. Below, we address the reviewer's concerns in detail.
>
> ##  1. Algorithm description (W1, W4)
>
> The reviewer noted that some aspects of our algorithm description in Section 3.2 were unclear. We have revised the paper to clarify the following points:
>
> **1.1 Computation of principal components (PCs):** Line 4 of Algorithm 1 performs an eigendecomposition of the matrix $\bar{\Theta}\in\mathbb{R}^{C\times C}$ as $\bar{\Theta}= \mathbf{V}\Sigma\mathbf{V}^\top$ and extracts the top $k\leq C$ eigenvectors $\mathbf{V}_k = [v_1,\dots, v_k]\in\mathbb{R}^{C\times k}$. Their corresponding eigenvalues form $\Sigma_k=\text{diag}(\sigma_1,\dots,\sigma_k)$. Since $\bar{\Theta}$ is *dual* to the covariance matrix $\bar{\mathbf{S}}\in\mathbb{R}^{P\times P}$, they share the same nonzero eigenvalues, and the PCs of $\bar{\mathbf{S}}$ can be obtained by lifting the PCs of  $\bar{\Theta}$ into $\mathbb{R}^P$ via
> $$
> \mathbf{U}_k:=\bar{\mathbf{G}}\mathbf{V}_k\Sigma_k^{-1/2}\in\mathbb{R}^{P\times k}.
> $$
> Line 5 of Algorithm 1 carries out this lifting procedure. The associated projection matrix is then
> $$
> \mathcal{P}:=\mathbf{U}_k\mathbf{U}_k^\top\in\mathbb{R}^{P\times P}.
> $$
> In practice, we never store $\mathcal{P}$ explicitly, since storing $\mathbf{U}_k$ is enough to compute the score.
>
> **1.2 Score function:** Our detection score $s(x) := ||\mathcal{P}\bar{\mathbf{g}}(x)||/||\bar{\mathbf{g}}(x)||$ (defined in line 9 of Algoritm 1) is not a reconstruction error. Rather, it is the cosine of the angle between $\bar{\mathbf{g}}(x)$ and its projection onto the PC subspace. While reconstruction error is standard in PCA-based methods, our choice aligns with the observations from Guan et al. (2023) that the angle is more predictive for OOD detection. It also provides a normalized, scale-invariant measure. We note that our theoretical framework in Section 4 is formulated for this score and formally justifies why it is meaningful for OOD detection.
>
> **1.3 Aggregation function:** We appreciate the reviewer pointing out the lack of clarity regarding how the final scalar output is formed. We now explicitly state that the model's scalar output is $f(x)=\max_{c}f^c(x)$, where $f^c(x)$ denotes the network's logit for class $c\in\{1,\dots,C\}$.
>
> All of these clarifications have been added as remarks directly after Algorithm 1 in the revision.
>
> ## 2. PCA-based baselines (W2, W5)
>
> We thank the reviewer for suggesting comparisons with additional PCA-based approaches, and agree that this strengthens our work. We incorporated them into the main evaluation tables in the revision. Specifically, we now include results for: (1) Revisited PCA (Guan et al., 2023), (2) Projected Gradients (Wu et al., 2024), and (3) Kernel PCA with a cosine-Gaussian kernel (CoRP) (Fang et al., 2024).
>
> We also looked into the reviewer's concern regarding GradOrth's (Behpour et al., 2023) performance, and identified that the method was not consistently given a data subset covering all classes for estimating its projection matrix. After addressing this, GradOrth's performance improved and became comparable with the other PCA-based methods.
>
> For ease of comparison, we include below an aggregated table reporting the average performance on each benchmark.
>
> |Method|CIFAR10 BiT-M|CIFAR10 TIMM |CIFAR100 BiT-M|CIFAR100 TIMM|ImageNet BiT-S|ImageNet BiT-M| Avg. FPR95|Avg. AUC|
> |----------------------------------|:--------------:|:-------------:|:----------------:|:---------------:|:----------------:|:----------------:|:------------------:|:---------------:|
> |Revisited PCA|24.95/91.04|30.59/92.60|30.14/92.57|60.40/85.60|44.83/87.36|64.43/74.89|**42.56**|**87.34**|
> |Proj. Grads|25.59/93.24|42.60/84.90|51.88/87.85|74.72/76.55|60.78/78.03|78.29/73.55|**55.64** |**82.35**|
> |GradOrth|73.95/79.74|40.42/91.97|31.79/92.89|68.84/85.50| 34.97/90.34|95.88/66.08|**57.64**|**84.42**|
> |CoRP|23.12/95.38|39.15/93.68|23.12/95.38|57.78/87.65|60.11/78.65|56.07/78.77|**43.23**|**88.25**|
> |GradPCA|20.15/95.96|36.82/92.85|27.87/94.01|67.78/86.04|36.62/90.22|66.22/82.10|**42.58**|**90.20**|
>
> Overall, these method show similar qualitative trends to GradPCA, performing well on pre-trained models. GradPCA achieves the highest global average performance. CoRP is also competitive, though unlike GradPCA, it is sensitive to hyperparameters (e.g., kernel bandwidth, RFF dimensionality) and requires tuning.
>
>
> ## 3. Consistency w.r.t. random seed (W3)
>
> We thank the reviewer for suggesting an evaluation of GradPCA sensitivity to the random seed. We have added this experiment to Appendix E.3. Concretely, we trained 10 VGG16 models on CIFAR10 with different seeds and evaluated GradPCA on each. The performance variation was mild, with AUC differences within 3% across models.
>
> ---
> We hope these clarifications and additions address the reviewer's main concerns.

---

> > ### Comment · Reviewer_zZXM · 2025-11-25
> >
> > Thanks for providing a detailed rebuttal. I appreciate the clarification and the supplemented results from the authors. All my comments are well addressed. I think the NTK perspective based on model parameter gradients is beneficial to OOD detection. I have raised my rating.

---

> > > ### Author Response · Authors · 2025-11-25
> > >
> > > Thank you for taking the time to consider our rebuttal and for the updated evaluation.

---

### Official Review · Reviewer_qt5N · 2025-10-28

**Soundness:** 2
**Presentation:** 3
**Contribution:** 2
**Rating:** 4
**Confidence:** 4

**Summary:**

This work proposes a new method for OOD detection by leveraging NTK alignment over a well-trained network, where both analytical discussions and  empirical evidence are both provided.

In general, the work is easy to follow and the NTK alignment technique is new to the task with different perspectives, even though the PCA technique and its operation on the gradients are not.

**Strengths:**

This method is easy to implement and shows good efficiency (no training). The empirical performance is on average good and partially near the state-of-the art performances. In particular, the involved NTK perspective is note fully investigated in this field, which may bring some new potential inspiring future work.

**Weaknesses:**

Some weakness (or points to be clarified) as below:

1. what if the labels of training data are not accessible? This method requires the labels/classes of training samples to construct the projection in PCA, but this is not really requested in other methods and might not be feasible in practical settings, especially considering data privacy. Could the authors make some discussions and remedies? Further, it would be good to mark out in the table whether this requirement  is taken in each compared method.

2. Up to section 3.2, why line 9 and line 10 in algorithm 1 can successfully enable OOD detection? In previous context, it mentions separability of features, which is yet still not direct to the exact rationale of such detection scores.

In line 149-155, the motivation is rather intuitive and based on existing work (He & Su, 2020). Could you also specify the motivation or evidence? Otherwise, it seems that this wok gives a new technique to do OOD with NTK and PCA, without a clear, strong and verifiable motivation/rationale to do so.

This is very important for the readability of this work and the clarification of its novelty.

3. Why not present discussions with (Guan et al., 2023)  and (Wu et al., 2024a) in the main context, but appendix, and why not present the empirical comparisons in the experiments? These works are highly related.

4. This work presents quite some discussions on the impact of "feature quality". It still seems unclear and not specific enough for the reviewer accessing the so-called "feature quality" quantitatively. Is it possible to have a clear definition  or metric? Please be more strictly technically precise.

**Questions:**

Please view the Weakness.

---

> ### Author Response · Authors · 2025-11-20
>
> We thank the reviewer for the careful evaluation of our work and for raising thoughtful concerns. Below, we address the identified weaknesses and outline the changes made in the revision.
> ## 1. Access to training data labels
> The reviewer correctly noted that our method requires access to ID labels during training. While this is a limitation, it is shared by widely used Mahalanobis OOD detectors and prototype-based approaches. Among recent PCA-based methods, the Projected Gradients approach of Wu et al. (2024) has the same requirement. A practical mitigation is to replace true labels with pseudo-labels or clusters obtained in an unsupervised manner. For example, Sehwag et al. (2021) [1] study an unsupervised variant of a Mahalanobis-based detector that estimates class structure through clustering in feature space. Similar strategies could be applied to GradPCA, and we view this as a promising direction for future work.
>
> We have added this discussion to the limitations section of the conclusion and marked the ID label requirement in Table 1. We thank the reviewer for highlighting this point.
>
> [1] Sehwag et al. "SSD: A unified framework for self-supervised outlier detection". ICLR, 2021.
> ## 2.1 Justification of the OOD score
>
> The reviewer asked why our score $s(x):=||\mathcal{P}\bar{\mathbf{g}}(x)||/||\bar{\mathbf{g}}(x)||$ can enable OOD detection. A formal justification is provided in Section 4, which develops a theoretical framework for spectral OOD detection. In particular, Theorem 4.1 shows that, in an idealized low-rank covariance setting, any score below 1 certifies OOD. Theorem 4.2 extends this result to settings with deviations from the perfect low-rank model and provides an explicit bound on the score.
>
> We emphasize that, although spectral and PCA-based OOD detectors are common in the literature, formal analyses explaining why such scores are meaningful for OOD detection are typically absent. Therefore, we view the theoretical results in Section 4 as an important contribution of this work.
>
> To make this clear earlier in the paper, we have added a remark on the score function (Remark 3.2) immediately after Algorithm 1 in the revision.
> ## 2.2 Motivation
> The reviewer requested additional motivation for using NTK alignment beyond the explanation in lines 149-155, which discusses how the NTK block structure relates to local elasticity. In the revision, we address this by including Figure 1, which visually illustrates the NTK block structure across several neural image classifiers.
> ## 3. Comparison with PCA-based detectors
> We thank the reviewer for suggesting a comparison with recent PCA-based detectors and agree that this strengthens our work. Following this suggestion, we have expanded our baselines to include: (1) Revisited PCA (Guan et al., 2023), (2) Projected Gradients (Wu et al., 2024), and (3) Kernel PCA with a cosine-Gaussian kernel (CoRP) (Fang et al., 2024).  These methods are now included in the main evaluation tables of the revision and discussed in Section 2, in addition to the appendix.
>
> For ease of presentation, we provide below the aggregated results for the two methods specifically mentioned by the reviewer.
>
> |Method|CIFAR10 BiT-M|CIFAR10 TIMM |CIFAR100 BiT-M|CIFAR100 TIMM|ImageNet BiT-S|ImageNet BiT-M| Avg. FPR95|Avg. AUC|
> |----------------------------------|:--------------:|:-------------:|:----------------:|:---------------:|:----------------:|:----------------:|:------------------:|:---------------:|
> |Revisited PCA|24.95/91.04|30.59/92.60|30.14/92.57|60.40/85.60|44.83/87.36|64.43/74.89|**42.56**|**87.34**|
> |Proj. Grads|25.59/93.24|42.60/84.90|51.88/87.85|74.72/76.55|60.78/78.03|78.29/73.55|**55.64** |**82.35**|
> |GradPCA|20.15/95.96|36.82/92.85|27.87/94.01|67.78/86.04|36.62/90.22|66.22/82.10|**42.58**|**90.20**|
> ## 4. Discussion on feature quality
> The reviewer asked whether it is possible to provide a formal and quantitative definition of "feature quality". In the introduction and in Section 3.3, we say that models have *general-purpose features* when they are pretrained on large-scale datasets and subsequently fine-tuned on the ID data. This usage follows transfer learning literature, where feature quality is reflected by how well pretrained models can be fine-tuned across a variety of downstream tasks. Quantifying "how general-purpose" such features are is inherently difficult, as the answer depends on the family of downstream tasks considered. This difficulty also mirrors a fundamental limitation of OOD detection: we never have access to the full diversity of possible OOD data.
>
> Importantly, our statements about "feature quality" should be understood in this concrete sense: pretrained models (on ImageNet-21k) versus models trained from scratch on the ID data. To avoid implying a more universal notion than we intend, we have revised Section 3.3 to explicitly restate this definition (lines 266-267).
>
> ---
> We hope that our responses clarify the reviewer's main concerns.

---

### Official Review · Reviewer_LCME · 2025-10-31

**Soundness:** 3
**Presentation:** 3
**Contribution:** 3
**Rating:** 4
**Confidence:** 4

**Summary:**

The paper uses NTK alignment theory (that neural networks are low-rank subspaces) to effectively detect OOD samples. GradPCA is an OOD detection framework that applies PCA to neural network gradients.The authors use NTK Alignment to make this computationally tractable.

**Strengths:**

The paper establishes strong theoretical motivation behind the GradPCA framework along with some reasoning as to why it works and when the OOD detectors can be thought of as reliable, along with complementing empirical results.

The proposed method is computationally tractable and therefore more practical. O(NP) to O(C).

Decent insights on feature quality explaining when regularity based methods outperform abnormality based methods and vice-versa.

Good empirical results.

**Weaknesses:**

How does GradPCA compare to the similar recent work [1] which also does PCA on the gradients? Would be good if the authors clarify the major differences and their contributions.

[1] Wu, Yingwen, et al. "Low-dimensional gradient helps out-of-distribution detection." IEEE Transactions on Pattern Analysis and Machine Intelligence (2024).


Only 3 models seem to be used in the experimentation. A greater diverse set might have introduced models that might not exactly follow the NTK alignment theory and cause GradPCA to degrade, and a brief insight into that would have been nice.


Sensitivity to parameter subset selection is important since the method is primed at exploiting the low-rankness of the neural network. So, just using the final hidden layer might be sub-optimal, and a brief insight on this would be helpful.


Ablation study as to how to approximate class-means (smaller batch-sizes) would also have been helpful as they are primary in the proposed algorithm.

**Questions:**

Please see weaknesses above.

---

> ### Author Response · Authors · 2025-11-20
>
> We thank the reviewer for their thoughtful comments. Below, we address each concern and outline the changes made in the revision.
>
> ## 1. Comparison with Projected Gradients (Wu et al., 2024)
>
> We thank the reviewer for highlighting the related work by Wu et al. (2024). Our paper discussed it in Appendix A.2 under "Gradient-based approaches". In the revision, we also include empirical comparisons with this method (Tables 1, 2, 4). For convenience, the aggregated results are shown in the following table.
> |Method|CIFAR10 BiT-M|CIFAR10 TIMM |CIFAR100 BiT-M|CIFAR100 TIMM|ImageNet BiT-S|ImageNet BiT-M| Avg. FPR95|Avg. AUC|
> |----------------------------------|:--------------:|:-------------:|:----------------:|:---------------:|:----------------:|:----------------:|:------------------:|:---------------:|
> |Proj. Grads|25.59/93.24|42.60/84.90|51.88/87.85|74.72/76.55|60.78/78.03|78.29/73.55|**55.64** |**82.35**|
> |GradPCA|20.15/95.96|36.82/92.85|27.87/94.01|67.78/86.04|36.62/90.22|66.22/82.10|**42.58**|**90.20**|
>
> On the methodological side, the differences between Wu et al. (2024) and GradPCA are as follows:
> * Wu et al. use the gradient class-means directly as projection directions, while GradPCA performs PCA on the gradient class-means.
> * Wu et al. additionally train a small *auxiliary network* (BatchNorm + Linear layer) on the projected gradients and use confidence-based OOD scores on its logits, while GradPCA uses the score $s(x):=||\mathcal{P}\bar{\mathbf{g}}(x)||/||\bar{\mathbf{g}}(x)||$ applied directly to the gradients.
>
> We highlight the following points:
> * **Theoretical foundations:** GradPCA is intentionally simple and analytically tractable, which enables the theoretical framework in Section 4, including the per-sample OOD certificate (Theorem 4.2). Obtaining comparable guarantees for Wu et al. is challenging due to the additional heuristics (training an auxiliary network and applying separate scoring functions).
> * **Computational cost:** Training the auxiliary network in Wu et al. can be costly on large datasets. Moreover, their method stores all $C$ gradient class-means in memory, while GradPCA typically retains fewer PCs due to the variance threshold $\epsilon$ (default $0.99$). Thus, GradPCA is more efficient in memory, training time, and inference.
>
> ## 2. Scope of models
> While our main benchmarks include two models for each ID dataset, and an additional ViT model (seven models in total), we agree that evaluating GradPCA under weaker NTK alignment is of interest. In the revision, we therefore added an experiment in which VGG16 models are trained on CIFAR10 with varying levels of label noise. This setup degrades the model's accuracy, weakens NTK alignment, and consequently reduces GradPCA's performance. The results are presented in Appendix E.11.
>
> ## 3. Parameter subset ablation
> We agree with the reviewer that GradPCA depends on the choice of parameter subset. In fact, our CIFAR10 results (Table 4) already included GradPCA applied to intermediate-layer parameters. Specifically, the 4th residual block (“block 4”) of BiT-M and the 3rd residual block (“block 3”) of the TIMM model. These variants outperform the default version, though at increased computational cost due to the larger parameter subsets, as discussed in Appendix E.1.
>
> In the revision, we include more extensive ablations of parameter subsets in Appendix E.4. These cover additional residual blocks for CIFAR10 (Table 8) and individual convolutional units for CIFAR100 (Table 9). The results confirm that certain intermediate layers closer to the network's output yield better performance.
>
> Prior work on NTK alignment has shown that alignment strength varies considerably across layers: early layers typically exhibit weaker alignment, deeper layers stronger alignment, and the maximum often occurs in intermediate layers (see Lou et al. (2022), Baratin et al. (2021)). This aligns with our observation that applying GradPCA to intermediate layers is often effective. However, the precise optimal subset does not transfer cleanly across models or datasets. Therefore, we use the last-layer parameters as the default, as this choice is computationally efficient and consistently offers strong performance.
>
> Finally, we note that understanding how NTK alignment and gradient representations evolve across layers remains an open problem in deep learning theory. Progress in this direction may guide more principled parameter subset selection for GradPCA.
>
> ## 4. Training data subset ablation
> We agree with the reviewer that it is important to assess how sensitive GradPCA is to the amount of data used to approximate the class-means. Accordingly, we have added an ablation in Appendix E.5 of the revision, where we train GradPCA on ImageNet subsets of varying sizes. Notably, the performance of GradPCA trained on only 10\% of the ID data is not worse than when using the full dataset.
>
> ---
> We hope these responses address the reviewer's main concerns.

---

### Official Review · Reviewer_vPae · 2025-10-31

**Soundness:** 4
**Presentation:** 3
**Contribution:** 3
**Rating:** 8
**Confidence:** 5

**Summary:**

This paper introduces GradPCA, a novel and principled method for Out-of-Distribution (OOD) detection. The core idea is to exploit the low-rank structure of neural network gradients, which the authors connect theoretically to the Neural Tangent Kernel (NTK) alignment phenomenon. The method applies Principal Component Analysis (PCA) to the class-means of the gradients to define a low-dimensional "in-distribution" subspace. At inference, inputs whose gradients fall outside this subspace are flagged as OOD.
The authors provide a theoretical framework for spectral OOD detection, including a sufficient condition that yields per-sample OOD certificates. Empirically, the paper demonstrates that GradPCA achieves highly consistent and competitive performance across several standard benchmarks (CIFAR-10, CIFAR-100, ImageNet). A significant additional contribution is a systematic analysis showing that OOD detector performance critically depends on feature quality (neural collapse property).

**Strengths:**

- Strong Theoretical Contribution: The paper offers a formal framework for spectral OOD detection in NNs. Theorem 4.1, which provides a "sufficient condition for spectral OOD detection" , is a strong theoretical result that provides a deterministic, per-sample OOD certificate.

- Excellent Empirical Results & Consistency: GradPCA achieves SOTA or near-SOTA results, but more importantly, it demonstrates the most consistent performance across all benchmarks. This directly addresses a major problem in the OOD field, where methods often fail erratically in different settings.

- Valuable Analysis of Feature Quality: The paper's distinction between "regularity-based" and "abnormality-based" detectors is insightful. The finding that their effectiveness is tied to whether features are pretrained (general-purpose) or not (task-specific) is a practical and important contribution that helps reconcile inconsistencies in prior work.

**Weaknesses:**

- Memory Scalability: The primary weakness is that the memory cost scales with the number of classes, $C$. The method stores $O(C)$ gradient vectors, which "can be costly for large C" like in ImageNet15. While the paper shows this is manageable (e.g., 7.5GB for ImageNet in the worst case, but often less  ), it could be a barrier for datasets with thousands or tens of thousands of classes.

- Core Assumption: The method's success relies on the assumption that NTK alignment provides a low-rank structure that effectively separates ID and OOD points in the gradient space . The authors note this "may not hold universally," though they rightly argue this assumption is explicit and empirically supported.

- Some missing references and comparisons that similarly leverage the gradient space:

Lee, Jinsol, et al. “Gradient-Based Adversarial and Out-of-Distribution Detection.” arXiv:2206.08255, arXiv, 4 July 2022. arXiv.org, http://arxiv.org/abs/2206.08255.

Sun, Jingbo, et al. “Gradient-Based Novelty Detection Boosted by Self-Supervised Binary Classification.” arXiv:2112.09815, arXiv, 17 Dec. 2021. arXiv.org, http://arxiv.org/abs/2112.09815.

ElAraby, Mostafa, et al. "GROOD: GRadient-Aware Out-of-Distribution Detection." Transactions on Machine Learning Research. https://arxiv.org/abs/2312.14427

**Questions:**

The paper mentions that GradPCA defaults to using only the last hidden layer parameters. It would be beneficial to include a more comprehensive ablation study on this choice. How does the method's performance vary when using gradients computed from the output layer as in GradNorm, or from specific intermediate blocks?

To further strengthen the empirical validation, the authors are encouraged to benchmark GradPCA within the OpenOOD framework. This would enable a standardized comparison and, more importantly, provide valuable insights into the method's performance on challenging near-OOD datasets versus far-OOD datasets.

For improved readability, especially for black-and-white printing, the authors should reconsider the highlighting scheme in the results tables. The current use of color to highlight the top-3 performers  can be confusing. A simpler, more standard convention, such as bolding the top-performing method and underlining the second, would significantly enhance clarity.

The paper's analysis of feature quality versus detector performance is a key contribution . To further explore this relationship, could the authors include an ablation study where feature quality is degraded in a more controlled manner? For instance, how do the "regularity-based" and "abnormality-based" detectors compare when the in-distribution (ID) training dataset is corrupted with increasing levels of label noise or input noise?

---

> ### Author Response · Authors · 2025-11-20
>
> We thank the reviewer for their thoughtful feedback and for the positive evaluation of our work. Below, we address the reviewer's questions and concerns and outline the changes made in the revision.
>
> ## Parameter subset ablation (Q1)
> We thank the reviewer for highlighting that GradPCA can be applied to arbitrary parameter subsets, and that performance may depend on this choice. Indeed, our CIFAR10 results (Table 4) already included variants of GradPCA applied to intermediate-layer parameters. Specifically, the 4th residual block (“block 4”) of BiT-M and the 3rd residual block (“block 3”) of the TIMM model. These variants outperform the default version, though at increased computational cost due to the larger parameter subsets, as discussed in Appendix E.1.
>
> However, we agree with the reviewer that a comprehensive ablation was lacking. Therefore, in the revision, we included more extensive ablations of parameter subsets in Appendix E.4. These cover additional residual blocks for CIFAR10 (Table 8) and individual convolutional units for CIFAR100 (Table 9). The results confirm that certain intermediate layers closer to the network's output yield better performance.
>
> We note that prior works on NTK alignment observed that alignment strength varies considerably across layers: early layers typically exhibit weaker alignment, deeper layers stronger alignment, and the maximum often occurs in intermediate layers (see Lou et al. (2022), Baratin et al. (2021)). This aligns with our observation that applying GradPCA to deeper intermediate layers is often effective,  while earlier layers yield poor performance. However, the precise optimal parameter subset does not transfer cleanly across models or datasets. Therefore, we use the last hidden layer parameters as the default, as this choice is computationally efficient and consistently offers strong performance.
>
> Finally, we note that understanding how NTK alignment and gradient representations evolve across layers remains an open problem in deep learning theory. Progress in this direction may guide more principled parameter subset selection for GradPCA.
>
> ## Additional benchmarks (Q2)
> We thank the reviewer for the suggestion to evaluate our method on OpenOOD, and in particular under the near-OOD setup. Since our method is implemented in JAX, directly integrating it into the OpenOOD pipeline is nontrivial within the review period. However, we fully agree that assessing near-OOD performance is important. To address this point, we have added a near-OOD experiment using CIFAR-10 (ID) vs. CIFAR-100 (OOD) in Appendix E.1.1 of the revision. As expected, this more challenging setting leads to a modest performance drop for GradPCA, as well as for the baselines, but no critical degradation is observed.
>
> ## Tables formatting (Q3)
> As suggested by the reviewer, we have updated the formatting of the tables in the revision. Specifically, we removed the color coding for the top-3 performing methods in the main evaluation tables and instead highlight the best performer in bold and the next two performers with underlining.
>
>
> ## Feature quality (Q4)
> We thank the reviewer for raising the question of evaluating GradPCA under controlled degradation of feature quality. In the revision, we have added experiments evaluating GradPCA on models trained with varying degrees of label noise. The results are presented in Appendix E.11. As expected, label noise degrades the model's accuracy, weakens NTK alignment, and consequently reduces GradPCA's performance. However, we note that poor classification performance is generally associated with degraded OOD detection performance in neural networks.
>
> At the same time, we are not aware of a principled way to degrade feature quality while preserving model accuracy. In this context, our comparison between pretrained and non-pretrained models offers a rare setting where classification performance is comparable, but the learned representations differ substantially. We view the identification and use of this contrasting setting as a contribution of our work.
>
> ## Memory scalability
> We agree with the reviewer that GradPCA incurs a memory overhead, which is a limitation shared by most spectral and regularity-based methods. We would like to highlight that we further analyze this overhead in Appendix E.7, where we report the number of principal components retained for ImageNet models under varying variance thresholds $\epsilon$. These results show that the number of stored vectors is typically much smaller than $C$, and can be reduced further with only minor performance degradation.
>
>
> ## Additional references
> We thank the reviewer for pointing us to additional gradient-based approaches. We have added discussion of these methods into Appendix A of the revision.
>
> ---
> We hope that these responses help clarify the reviewer's questions.

---

### Official Review · Reviewer_U4mW · 2025-10-31

**Soundness:** 3
**Presentation:** 3
**Contribution:** 2
**Rating:** 6
**Confidence:** 4

**Summary:**

The paper proposes GradPCA that applies PCA to class-mean gradients of a trained neural network. The approach is motivated by Neural Tangent Kernel (NTK) alignment, which empirically yields a low-rank, approximately block-diagonal structure in the gradient covariance of well-trained models. GradPCA formalizes this link between NTK alignment and spectral OOD detection, derives sufficient and necessary conditions for detection guarantees, and demonstrates consistent, near–state-of-the-art performance across CIFAR-10/100 and ImageNet under pretrained and non-pretrained settings.

**Strengths:**

Overall, this is a well-written and sophisticated paper with clear motivation, background, and rationale.  The writing is made to be accessible to a general deep-learning audience.

The paper has a clear theoretical grounding that connects OOD detection to NTK alignment and covariance low-rank structure, offering a mathematically elegant view. The method essentially captures a low-rank representation of the NTK kernel / gradient covariance matrix. Kernel-based reasoning is well-founded; kernel and spectral methods have a strong theoretical pedigree and proven reliability for measuring similarity in DL to model complex latent spaces of features. The approach aligns conceptually with sparse / low-rank matrix representation theory, which is also a mature and robust field.
Experiments have been conducted on multiple benchmark datasets. Empirically results are generally sound and demonstrate performance suggested by the theory.

**Weaknesses:**

The choice of NTK, while natural given the theoretical link, is not unique. Other kernels (e.g., Fisher information, feature-space kernels) could also exhibit similar low-rank behavior, so the generality of the method is not fully demonstrated. The paper would benefit from a clearer articulation of why the NTK is the most appropriate or insightful kernel for connecting spectral structure with OOD behavior. In other words, it needs to “ring a bell” by making the NTK–OOD link feel both necessary and intuitively strong.

Experimental coverage is confined to image classification. It remains unclear whether the same assumptions and empirical stability extend to other domains such as text, time series, etc. While full experiments are not required, a discussion of applicability and limitations across modalities would strengthen the paper.

The theoretical results rely on simplified assumptions (rank-C covariance matrix, small residual, etc.) that may not strictly hold in real neural networks. What if some of the classes have similar, or even partially overlapping feature semantics?

Related comments are provided in the questions section below.

**Questions:**

Could the method be extended to characterize OOD detection in terms of Type I and II errors, rather than a hard binary boundary? Specifically, how would the approach behave if in-distribution (ID) and OOD supports overlap, as might occur with visually similar classes?

As a related question, the theoretical assumptions (low-rank structure, small residual \xi) may not hold when classes are mixed or poorly separated. Have the authors tested how sensitive GradPCA is to such violations?

How is the kernel chosen? Could alternative kernels (e.g., Fisher, feature-space, or adaptive kernels) improve detection? Is the NTK selection fixed or adaptive to data/model characteristics?

For the covariance (aka $FF^\top$) matrix, are gradients centered before computing PCA? Clarification is needed on whether centering affects the eigenstructure and stability of the projection.

---

> ### Author Response · Authors · 2025-11-20
>
> We thank the reviewer for their thoughtful comments and the positive evaluation of our work. We address the raised concerns below.
>
> ## Support-based vs. likelihood-based OOD detection (Q1)
> We thank the reviewer for raising the question about the relationship between our results and Type I/II errors in the overlapping supports setting. This connects directly to our discussion in Appendix B.1, where we contrast the *support-based* definition of OOD detection proposed in our paper with the *likelihood-based* definition that is common in the literature.
>
> The core issue is the following: for a given input $x$, the optimal likelihood-based detector (Eq. (14)) requires comparing the unknown likelihoods $\mu_{in}(x)$ and $\mu_{ood}(x)$. In particular, the decision on whether $x$ is OOD depends on the (unknown and arbitrary) value of $\mu_{ood}(x)$. However, whether a network will correctly classify $x$ cannot depend on how an external OOD distribution assigns likelihood to that point. This mismatch creates conceptual difficulties when relating likelihood-based definitions to the model's domain of competence.
>
> Our support-based perspective avoids this dependence on $\mu_{ood}$ and provides an optimal detector that depends only on the support of the ID distribution. This aligns more naturally with the goal of OOD detection: identifying inputs on which the model is likely to fail. This viewpoint also enables us to derive formal OOD certificates (Theorems 4.1 and 4.2), which characterize precisely when the Type I error is zero.
>
> To our knowledge, analogous results cannot be derived under the likelihood-based perspective, particularly when supports overlap, which may help explain the scarcity of theoretical guarantees in the OOD detection literature. We therefore view the support-based formulation, and the theoretical guarantees it enables, as a key conceptual contribution of our work.
>
> ## GradPCA under violated assumptions (Q2)
> We agree with the reviewer that it is important to evaluate GradPCA's behavior when the NTK block-structure assumption does not hold. In the revision, we have added experiments evaluating GradPCA on models trained with varying degrees of label noise. The results are presented in Appendix E.11. As expected, label noise degrades the model's accuracy, weakens NTK alignment, and consequently reduces GradPCA's performance. However, we note that poor classification performance is generally associated with degraded OOD detection performance in neural networks.
>
> At the same time, we are not aware of a controlled way to break the NTK block structure *without simultaneously harming model accuracy*. In practice, well-trained image classifiers exhibit NTK block structure, which makes it challenging to isolate violations of this assumption without also degrading the underlying model.
>
> ## Choice of the NTK kernel (Q3)
> We agree with the reviewer that the kernel choice in our method is not unique. This is precisely why the theoretical framework in Section 4 is formulated in terms of a general feature map $h$, rather than the specific gradient map used by GradPCA.
>
> Our choice of the NTK is motivated by a substantial body of work on NTK alignment, which shows that the NTK develops a pronounced low-rank structure across a wide range of practical deep learning settings (e.g., Atanasov et al. (2022), Baratin et al. (2021), Shan & Bordelon (2021), He & Su (2020), Seleznova et al. (2023)). These results support the NTK as an effective choice for applying our framework. In addition, because the NTK is a central object in deep learning theory, progress in understanding its behavior is likely to directly inform and strengthen OOD detection methods such as ours.
>
> In the revision, we also include empirical comparisons with several other spectral OOD detection methods: Revisited PCA (Guan et al., 2023), Projected Gradients (Wu et al., 2024), and Kernel PCA with a cosine-Gaussian kernel (CoRP) (Fang et al., 2024). These comparisons provide additional perspective on how other commonly used kernels and feature maps perform in spectral OOD detection.
>
> That said, we agree that other kernels with useful low-rank properties may exist and could be incorporated into our method. Identifying such kernels is an important direction for future work, both for spectral OOD detection and for deep learning theory more broadly, as we also note in the conclusion of our paper.
>
> ## Centering in PCA (Q4)
> The reviewer asked whether gradients are centered before applying PCA. As specified in lines 3 and 8 of Algorithm 1, we do center the gradients. Centering is an essential step in PCA: it ensures that PCA captures directions of maximal variance in the data. Without centering, the first principal component would align with the mean vector, which is not consistent with the objective of PCA.
>
> ---
> We hope that these responses help clarify the reviewer's questions.

---

### Author Response · Authors · 2025-11-20

We thank all the reviewers for evaluating our work and the AC for overseeing the reviewing process. We appreciate that the reviewers have recognized both the theoretical contributions of our work, including the formal framework for spectral OOD detection, and the practicality and effectiveness of the proposed method.

Below, we summarize the main changes incorporated in the revision in response to the reviewers' feedback:

* **Expanded baselines:** As suggested by the reviewers, we added several recent PCA-based OOD detection methods to the main evaluation tables (Tables 1, 2, and 4): (1) Revisited PCA (Guan et al., 2023), (2) Projected Gradients (Wu et al., 2024), and (3) Kernel PCA with a cosine–Gaussian kernel (CoRP) (Fang et al., 2024).
* **Additional ablations:** The revision includes the following new ablations requested by the reviewers: (1) parameter subset selection for GradPCA (Appendix E.4), (2) sensitivity to the fraction of ID data used to compute gradient class-means (Appendix E.5).
* **Robustness to random seeds:** Appendix E.3 now includes experiments demonstrating GradPCA's consistency across training runs with different random seeds, extending our discussion on consistency.
* **Impact of label noise:** Appendix E.11 presents experiments evaluating how varying degrees of label noise during training affect GradPCA performance, which provides an example of a setting where the method's assumptions are violated.
* **Clarified algorithm description and motivation:** We expanded Section 3.2 to give a clearer account of how principal components are computed and to further motivate our choice of score function. We also supplemented our discussion of NTK alignment with a new Figure 1 illustrating the NTK block structure.
* **Expanded discussion of limitations:** In the conclusion, we now explicitly state that our method requires ID labels during training and discuss possible mitigations. We also clarify that our formulation is specific to classification settings.
* **Improved table formatting:** To ensure readability in black-and-white printing, we removed color highlighting in the benchmark tables and now highlight top performers using boldface and underlining.

We believe that these revisions substantially strengthen the paper, and we hope they address the majority of the reviewers' concerns.

---

### Meta-Review · Area_Chair_uDXg · 2026-01-12

**Summary:**

This paper proposes GradPCA, an out-of-distribution detection method that applies PCA to class-mean gradients to exploit the low-rank structure induced by Neural Tangent Kernel alignment. The authors provide a theoretical analysis linking the spectral properties of gradients and features to effective OOD detection and show how representation quality influences detector performance. Extensive experiments on standard image benchmarks demonstrate that GradPCA achieves more consistent performance than existing baselines. Overall, the work offers both a practical method and a principled perspective on spectral approaches to OOD detection.

During the rebuttal, the authors have addressed the reviewers’ concerns well. In addition to the existing suggestions, we recommend that the authors include one more baseline [1], as the neural collapse phenomenon in feature space is closely related to NTK behavior in gradient space, and [1] also reports consistent performance across multiple benchmarks—one of the key claims of this work. A direct comparison between these two methods would further strengthen the evaluation. Taken together, we recommend accepting this work.

[1] Detecting Out-of-Distribution through the Lens of Neural Collapse. CVPR 2025.

**Reviewer Concerns:**

All addressed.

**Reviewer Scores:**

Reviewer U4mW (score: 6) would potentially maintain the score.

Reviewer vPae (score: 8) would potentially maintain the score.

Reviewer LCME (score: 4) would potentially increase the score as the concerns are addressed.

Reviewer qt5N (score: 4) would potentially increase the score as the concerns are addressed.

Reviewer zZXM (score: 6) confirmed to maintain the score.

---

### Decision · Program_Chairs · 2026-01-26

Accept (Poster)